# Dynamic control of hippocampal spatial coding resolution by local visual cues

Romain Bourboulou[1†], Geoffrey Marti[1†], François-Xavier Michon[1], Elissa El Feghaly[1], Morgane Nouguier[1], David Robbe[1], Julie Koenig[1,2‡*], Jerome Epsztein[1‡*]

[1]Institute of Neurobiology of the Mediterranean Sea (INMED), Turing Center for Living Systems, Aix-Marseille Université, INSERM, Marseille, France; [2]Institut Universitaire de France, Paris, France

**Abstract** The ability to flexibly navigate an environment relies on a hippocampal-dependent cognitive map. External space can be internally mapped at different spatial resolutions. However, whether hippocampal spatial coding resolution can rapidly adapt to local features of an environment remains unclear. To explore this possibility, we recorded the firing of hippocampal neurons in mice navigating virtual reality environments, embedding or not local visual cues (virtual 3D objects) in specific locations. Virtual objects enhanced spatial coding resolution in their vicinity with a higher proportion of place cells, smaller place fields, increased spatial selectivity and stability. This effect was highly dynamic upon objects manipulations. Objects also improved temporal coding resolution through improved theta phase precession and theta timescale spike coordination. We propose that the fast adaptation of hippocampal spatial coding resolution to local features of an environment could be relevant for large-scale navigation.
DOI: https://doi.org/10.7554/eLife.44487.001

**\*For correspondence:**
julie.koenig@inserm.fr (JK);
jerome.epsztein@inserm.fr (JE)

[†]These authors contributed equally to this work
[‡]These authors also contributed equally to this work

**Competing interests:** The authors declare that no competing interests exist.

## Introduction

Animals can flexibly navigate their environment. In mammals such as rodents and humans, this ability is thought to rely on an internal cognitive map (*Tolman, 1948*; *O'Keefe and Nadel, 1978*; *Epstein et al., 2017*). When animals move in their environment, hippocampal place cells fire in specific locations (their place fields) and this spatial tuning is believed to provide a neuronal substrate to the cognitive map. To be useful for navigation, such internal representation should be properly oriented and positioned in reference to the external world (*Knierim and Hamilton, 2011*). A dynamic control of hippocampal spatial coding resolution between different regions of the same environment could be important for spatial navigation (*Geva-Sagiv et al., 2015*). Wild animals, including rodents, often travel kilometers away from their home through empty space to specific food locations (*Taylor, 1978*). Mapping all traveled space at similar spatial resolution would require a huge neuronal and computational investment. Alternatively, mapping different parts of the same environment at different spatial resolutions could be advantageous.

In rodents, hippocampal place cell coding can vary both qualitatively and quantitatively. Qualitatively, place cells can code more or less accurately for space depending on several behavioral/experimental manipulations such as passive vs active movements (*Terrazas et al., 2005*) or light vs dark conditions (*Lee et al., 2012*). A better accuracy is generally associated with decreased place field size, increased spatial and temporal place field stability upon repeated visits of the same location and low out-of-field firing rate. Quantitatively, the number of spatially selective cells and place fields' density can increase globally in the presence of objects (*Burke et al., 2011*) and locally near rewarded locations (*O'Keefe and Conway, 1978*; *Hollup et al., 2001*; *Dupret et al., 2010*; *Danielson et al., 2016*; *Gauthier and Tank, 2018*; *Sato et al., 2018*), salient sensory cues

(*Wiener et al., 1989*; *Hetherington and Shapiro, 1997*; *Sato et al., 2018*) or connecting parts in multi-compartment environments (*Spiers et al., 2015*). Whether these overrepresentations correspond to position coding at higher spatial resolution (i.e. the resolution of the 'where' information) or the coding of nonspatial information associated with these particular locations (also referred to as 'what' information) is, however, difficult to disentangle. If they would represent increased spatial resolution, then place fields should not only be more numerous but they should also more accurately code for space in terms of spatial selectivity, spatial information content and stability. Furthermore, in the context of navigation, spatial coding resolution should be rapidly adjustable within different parts of the same environment or upon specific experimental manipulations. Finally, improved spatial coding resolution should extend to the temporal coding domain.

The factors controlling spatial coding resolution are still poorly understood. While distal visual cues play an important role in map orientation and environmental boundaries in map anchoring (*O'Keefe and Burgess, 1996*; *Knierim and Rao, 2003*; *Knierim and Hamilton, 2011*), local sensory cues, with a high sensory resolution, could be instrumental in setting spatial coding resolution (*Hartley et al., 2000*; *Strösslin et al., 2005*; *Barry et al., 2006*; *Sheynikhovich et al., 2009*; *Geva-Sagiv et al., 2015*). Here, we took advantage of virtual reality (*Hölscher et al., 2005*; *Harvey et al., 2009*; *Youngstrom and Strowbridge, 2012*; *Ravassard et al., 2013*; *Aronov and Tank, 2014*; *Cohen et al., 2017*; *Thurley and Ayaz, 2017*; *Gauthier and Tank, 2018*) to specifically control and quickly manipulate local sensory cues and test their impact on hippocampal spatial coding resolution. We recorded a large number of hippocampal cells in area CA1 to be able to use decoding strategies to decipher the functional impact of the changes observed. Our results are consistent with a rapid adaptation of hippocampal spatial coding resolution to local features of the environment. We propose that this mechanism could be important for large-scale navigation.

## Results

### Effects of local visual cues on spatial coding resolution

To investigate the effect of local visual cues on hippocampal coding resolution, head-fixed mice were trained to run on a wheel and to shuttle back and forth on a 2 m-long virtual linear track to collect liquid rewards at its extremities (*Figure 1A*). The lateral walls of the virtual track displayed distinct visual patterns to provide directional information. To investigate the contribution of local cues to hippocampal spatial representation, mice were trained either in the presence or absence of 3D visual cues (hereafter called virtual objects; Objects Track, OT: n = 3 mice; No Objects Track, ØT: n = 3 mice), which were virtually positioned on the floor of the track between the animal trajectory and the walls (*Figure 1B*). The running wheel forced the animals to run in a unidirectional manner so that they repetitively ran along the virtual objects without the possibility to orient toward them or explore them with any sensory modality but vision. Animals received a reward (sucrose in water 5%) each time they reached one of the extremities of the linear track. After licking, the mice were 'teleported' in the same position but facing the opposite direction of the track (*Figure 1C*), allowing them to run back and forth in the same environment. Once animals reached a stable and proficient behavior (at least one reward/min during a 60-min-long session), we recorded spiking activity in the pyramidal cell layer of the CA1 hippocampal region using either 4-shanks or 8-shanks silicon probes (*Figure 1A*, *Figure 1—figure supplement 1*) in the right and/or left hemispheres over the course of 2–3 days. A total of 1021 neurons were recorded in the CA1 pyramidal cell layer in OT and ØT (*Supplementary file 1*). Mice trained in ØT performed the task with similar proficiency than mice trained in OT, as shown by similar rate of reward collections (ØT: 1.86 ± 0.31 rewards/minute, n = 9 sessions in three mice; OT: 1.44 ± 0.12 rewards/minute, n = 8 sessions in three mice; $Z = 0.52$, p=0.59, two-tailed Wilcoxon rank sum (WRS) test; all values expressed as mean ±SEM) and average running speed (ØT: 14.1 ± 2.12 cm/s, n = 9 recording sessions in three mice; OT: 15.3 ± 1.28 cm/s, n = 8 recording sessions in three mice; $t_{15} = -0.47$, p=0.64, two-tailed unpaired *t*-test).

We first assessed possible effects of local visual cues on overall hippocampal excitability by comparing the percentage of track-active putative pyramidal cells among all recorded cells in ØT and OT. The percentage of track active cells was comparable between the track without and with virtual objects (ØT: 66.4 ± 5.8%, n = 7 sessions in three mice; OT: 54.6 ± 4.8%, n = 8 sessions in three mice; $t_{13} = 1.58$, p=0.14, two-tailed unpaired *t*-test; *Figure 1D*). We next started to assess spatial coding

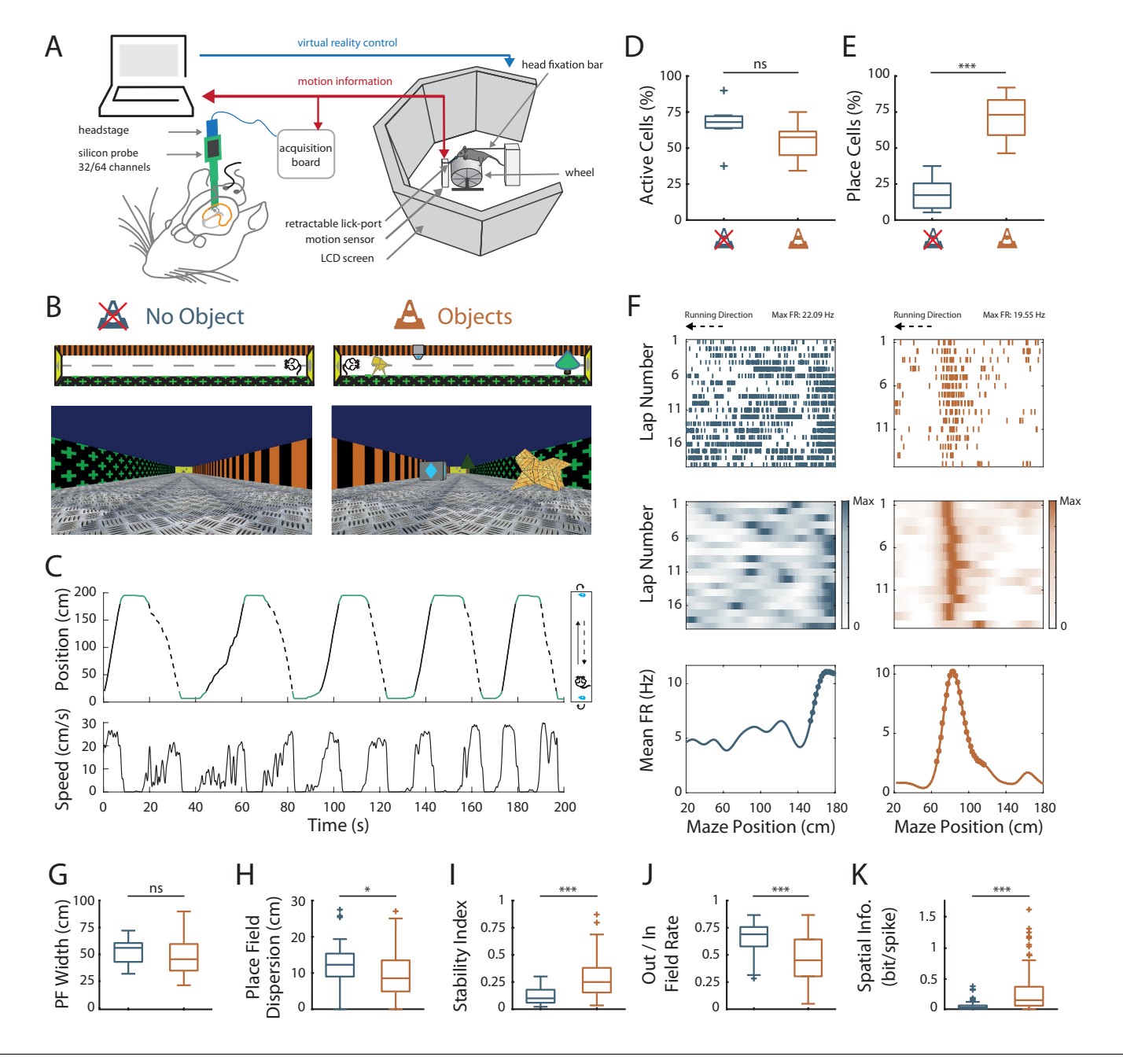

**Figure 1.** Effects of local visual cues on spatial coding resolution . (**A**) Schema of the virtual reality set up. The mouse is head-fixed and located on a wheel surrounded by LCD screens where a virtual environment is displayed. (**B**) Top and first person views of virtual linear tracks used. Left: track without objects (ØT) and right: track with virtual 3D objects (OT). (**C**) Top: Animal's position in the virtual track as a function of time. Green lines indicate times when animal was in a reward zone location. These locations were not considered for further analysis. Solid and dotted black lines indicate back and forth trials respectively. Top view of animal in the maze is depicted on the right. Arrows indicate teleportation in the same position but facing opposite direction after reward consumption. Bottom: Animal's speed as a function of time. (**D,E**) Box plots of the percentage of active cells (**D**) and place cells (**E**) in the maze without (blue) and with (orange) objects (same color code throughout the figures). (**F**) Spike raster plots (top) and color-coded firing rate map (middle) for successive trials in one direction (arrow) as a function of the position in the maze. Bottom: corresponding mean firing rate by positions. Dots indicate positions of the detected place field (see Materials and methods). (**G–K**) Box plots of the place field width (**G**), the place field dispersion (**H**), the stability index (**I**), the out/in field firing rate (**J**) and the spatial information (**K**). For box plots in this and subsequent figures, box extends from the first (Q1) to the third quartile (Q3) with the band inside showing the median and the extremities of the whiskers include values greater than Q1-1.5*(Q3-Q1) and smaller than Q3 +1.5*(Q3-Q1).

DOI: https://doi.org/10.7554/eLife.44487.002

*Figure 1 continued on next page*

*Figure 1 continued*

The following source data and figure supplements are available for figure 1:

**Source data 1.** Source data for *Figure 1*.
DOI: https://doi.org/10.7554/eLife.44487.005
**Figure supplement 1.** Histology and spike sorting.
DOI: https://doi.org/10.7554/eLife.44487.003
**Figure supplement 2.** Effects of local visual cues on spatial coding resolution across different recording sessions.
DOI: https://doi.org/10.7554/eLife.44487.004

resolution by comparing the proportion of place cells among active cells in the presence and absence of local visual cues. While only 19% of track active cells had at least one place field (place cells) in the empty track (n = 48 place cells), 71% of track active cells were place cells when virtual objects were present (n = 193 place cells; $t_{13}$ = −7.3, p<$10^{-5}$, two-tailed unpaired *t*-test; *Figure 1E*). In ØT, place fields were relatively sparse in the middle of the track with a large proportion of them aligned either to the beginning or to the end of the track (End-Track fields: 53.1 ± 9.95%, n = 7 sessions in three mice; *Figure 2A*). In the maze with objects, however, the majority of fields were located in the central part of the track (On-Track fields: 79 ± 3.52%; n = 8 sessions in three mice; $Z$ = 2.84, p=0.0045, two-tailed WRS test; *Figure 2A*). These results indicate that local visual cues can strongly increase the proportion of place cells among active cells notably to code the central part of the maze. Another factor influencing spatial resolution is place field size. There was a small, non-significant, tendency for place field width (calculated on complete fields) to be lower in the track with objects (ØT: 51.5 ± 3.33 cm, n = 15 place fields; OT: 48.7 ± 1.29 cm, n = 157 place fields; $Z$ = 0.93, p=0.35, two-tailed WRS test; *Figure 1G*), in agreement with a higher spatial coding resolution. The size of place fields based on single-trial detection was also not significantly different between the two conditions (ØT: 34.4 ± 1.2 cm, n = 15 place fields; OT: 34.2 ± 0.47 cm, n = 156 place fields; $Z$ = 0.51, p=0.61, two-tailed WRS test). On the other hand, the spatial dispersion of single-trial detected place fields was significantly reduced in the presence of 3D objects (ØT: 11.9 ± 0.90 cm, n = 48 place cells; OT: 9.70 ± 0.44 cm, n = 193 place cells; $Z$ = 2.56, p=0.01, two-tailed WRS test; *Figure 1H*). To further assess inter-trial place field stability, independently from place field detection, we calculated a stability index (based on spatial correlations between all pairs of firing rate vectors, see Materials and methods section). This stability index was significantly lower in the track without objects (ØT: 0.12 ± 0.01, n = 48 place cells; OT: 0.28 ± 0.01, n = 193 place cells; $Z$ = −6.64, p<$10^{-10}$, two-tailed WRS test; *Figure 1I*). Altogether, these results demonstrate that local visual cues can improve inter-trial spatial and temporal stability.

An increase in spatial coding resolution would also be associated with higher spatial selectivity and information content. Spatial selectivity was assessed by comparing the in-field versus out-of-field firing rates (i.e. signal-to-noise ratio) for place fields recorded in OT and ØT. In the track without objects, place cells increased their firing rate inside the place field (7.44 ± 0.75 Hz, n = 48 place cells) but also discharged at high rate outside the field (5.23 ± 0.62 Hz; *Figure 1F and J*; ratio: 0.65 ± 0.02). In comparison, place cells recorded in the track with objects had comparable firing rates inside the place field (6.80 ± 0.43 Hz, n = 193 place cells; $Z$ = 1.5, p=0.13, two-tailed WRS test) but fired significantly less outside the field (3.79 ± 0.34 Hz; ratio: 0.46 ± 0.01; *Figure 1F and J*; $Z$ = 5.48, p<$10^{-7}$, two-tailed WRS test). Accordingly, spatial information (in bit/spike), a measure independent of place fields' detection (*Skaggs et al., 1993*) was very low in the track without objects (0.06 ± 0.01 bit/spike, n = 48 place cells) and significantly higher in the presence of objects (0.25 ± 0.02 bit/spike, n = 193 place cells; $Z$ = −5.67, p<$10^{-7}$, two-tailed WRS test; *Figure 1K*). Similar results were obtained with a different method to estimate spatial information content based on the original mutual information metric with a normalization to correct possible bias due to differences in basal firing rates between conditions (*Souza et al., 2018*) (ØT: 1.67 ± 0.21, n = 48 place cells; OT: 5.62 ± 0.29, n = 193 place cells; $Z$ = −7.57, p<$10^{-13}$, two-tailed WRS test). The effects of objects on spatial coding resolution were also observed when comparisons were performed across recording sessions (*Figure 1—figure supplement 2* and *Supplementary file 2*).

Altogether these results indicate that local visual cues can strongly enhance the proportion of place cells among active cells but also place cell's coding accuracy in agreement with an improved spatial coding resolution.

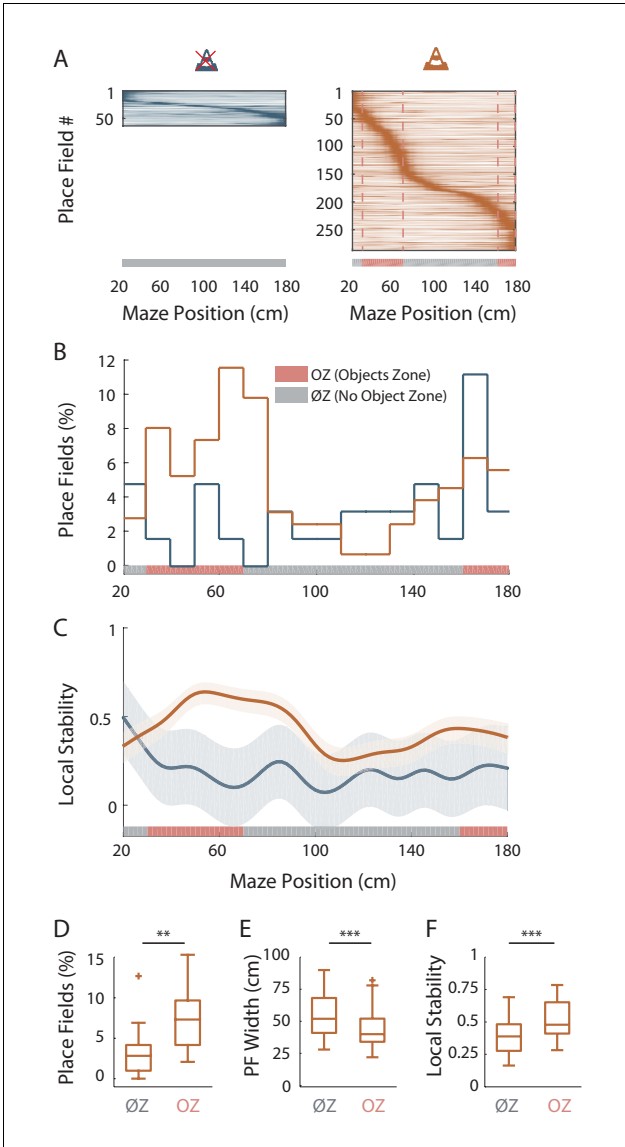

**Figure 2.** Virtual 3D objects improve spatial coding resolution locally. (**A**) Color-coded mean firing rate maps of all place fields recorded in the maze without objects (left) or with objects (right). The color codes for the intensity of the bin's mean firing rate normalized on the maximal mean firing rate (peak rate) in the recording session. The place cells are ordered according to the position of their peak rate in the track (reward zones excluded). Bottom: The tracks were divided into Objects Zones (OZ, in red on the x-axis) around the objects and No Object Zones (ØZ, in grey on the x-axis) deprived of objects. Red dotted lines depict the boundaries of the OZ in the track with objects. (**B**) Percentage of On-Track place fields at each spatial bin (10 cm) in the maze with (orange line) and without objects (blue line). (**C**) Mean local stability index (solid lines)±SEM (shaded bands) for place cells with On-Track fields at each spatial bin in the track with (orange) or without (blue) objects. (**D–F**) Box plots depicting the mean percentage of place fields per spatial bin (**D**), the place field width (**E**) and the local stability index (**F**) in OZ and ØZ in the maze with objects. *Figure 2—source data 1*. Source data for *Figure 2*.
DOI: https://doi.org/10.7554/eLife.44487.006

The following source data and figure supplement are available for figure 2:

**Source data 1.** Source data for *Figure 2*.
DOI: https://doi.org/10.7554/eLife.44487.008

**Figure supplement 1.** Virtual 3D objects improve spatial coding resolution locally across different recording sessions.
DOI: https://doi.org/10.7554/eLife.44487.007

## Virtual 3D objects improve spatial coding resolution locally

We then wondered whether spatial resolution could be adjusted locally, within the same environment. To address this question, we focused our analysis on On-Track fields recorded in the OT. We first noticed that the distribution of these fields was non-uniform (p=0.017, test of non-uniformity). To quantify more precisely this effect, we divided the linear track in Objects Zones (OZ) and No Objects Zones (ØZ), depending if a given track zone contained an object or not, respectively (*Figure 2A*, right). The density of place fields was significantly higher in OZ (OZ: 7.31 ± 1.09%/10 cm, n = 12 spatial bins of 10 cm, six in each direction; ØZ: 3.28 ± 0.65%/10 cm, n = 20 spatial bins of 10 cm, 10 in each direction; $t_{30}$ = −3.38, p=0.002, two-tailed unpaired $t$-test; *Figure 2B and D*). Furthermore, in the maze with objects, place fields were significantly smaller in OZ (42.3 ± 1.43 cm, n = 77 fields) compared to ØZ (54.8 ± 1.89 cm, n = 80 fields; $Z$ = 4.60, p<$10^{-5}$, two-tailed WRS test; *Figure 2E*). Accordingly, place field dispersion was also significantly reduced in OZ (8.33 ± 0.50 cm, n = 130 fields) compared to ØZ (11.8 ± 0.71 cm, n = 90 fields; $Z$ = 3.90, p<$10^{-4}$, two-tailed WRS test). A local stability index (see Materials and methods section) was significantly increased in OZ (0.52 ± 0.02, n = 60 bins of 2 cm, 30 in each direction) compared to ØZ (0.39 ± 0.01, n = 100 bins of 2 cm, 50 in each direction; $Z$ = −5.21, p<$10^{-6}$, two-tailed WRS test; *Figure 2C and F*). Spatial information was also significantly higher in OZ (0.32 ± 0.03 bit/spike, n = 130 fields) compared to ØZ (0.20 ± 0.03 bit/spike, n = 90 fields; $Z$ = −2.16, p=0.03, two-tailed WRS test). Finally, we found no significant difference in the out-of-field versus in-field firing ratio between fields located in OZ or ØZ (OZ: 0.46 ± 0.02, n = 130 fields; ØZ: 0.49 ± 0.02, n = 90 fields; $Z$ = 1.03, p=0.30, two-tailed unpaired $t$ test). The local effects of objects on spatial coding resolution were also observed when comparisons were performed across recording sessions (*Figure 2—figure supplement 1*).

These results indicate that 3D objects can locally improve spatial coding resolution through a local increase in place field number, a local reduction in place field size, a higher local stability and spatial information content while their effect on the out-of-field versus in-field firing ratio is more global.

We next wondered whether similar local effects on spatial coding resolution could be observed in ØT. In this track, place fields were also non-uniformly distributed (p=0; test of non-uniformity) with a higher density of fields at the ends of the track (i.e. End-Track fields; *Figure 2A*). However, we found no significant difference between End-Track and On-Track fields in terms of out-of-field versus in-field firing ratio (End-Track: 0.65 ± 0.02, n = 32 fields; On-Track: 0.62 ± 0.03, n = 31 fields; $Z$ = 0.21, p=0.83, two-tailed WRS test) and stability (End-Track: 0.17 ± 0.01, n = 32 fields; On-Track: 0.15 ± 0.02, n = 31 fields; $t_{61}$ = 1.14, p=0.26, two-tailed unpaired $t$-test). Spatial information was low for both types of fields but paradoxically lower for End-Track fields (End-Track: 0.04 ± 0.01 bit/spike, n = 32 fields; On-Track: 0.1 ± 0.02 bit/spike, n = 31 fields; $Z$ = −2.66, p=0.008, two-tailed WRS test). We conclude that overrepresentation of the ends of the ØT is not associated with increased spatial coding accuracy and is unlikely to represent increased spatial coding resolution at these locations.

## Effect of local visual cues on spatial coding resolution at the population level

The results so far suggest that hippocampal spatial coding resolution can be locally adjusted. To assess this at the population level, we next performed position-decoding analysis (*Brown et al., 1998*; *Zhang et al., 1998*) (*Figure 3A*). We used the spike trains from all pyramidal cells recorded (i.e. both the spatially modulated and nonspatially modulated cells) and compared decoded positions with actual positions of the animal in the virtual linear tracks. Overall, the effect of objects on hippocampal spatial coding was obvious because the decoding error across trials was nearly twofold larger in the track without objects compared to the track with objects (ØT: 46.3 ± 0.73 cm, n = 180 trials; OT: 27.1 ± 0.94 cm, n = 249 trials; $Z$ = 13.6, p<$10^{-36}$, two-tailed WRS test; *Figure 3A and B*). Accordingly, the decoding accuracy (*van der Meer et al., 2010*) was three fold lower in the empty track compared to the track with objects (ØT: 0.017 ± 3.8×$10^{-4}$, n = 180 trials; OT: 0.048 ± 1.49×$10^{-3}$, n = 249 trials; chance level 0.01; $Z$ = −15.68, p<$10^{-54}$, two-tailed WRS test; *Figure 3A and C*). In both cases, downsampling was performed to equalize the number of cells used for decoding between the two conditions (20 active cells). The effects of objects on population coding accuracy were also observed when comparisons were performed across recording sessions (*Figure 3—figure supplement 1*).

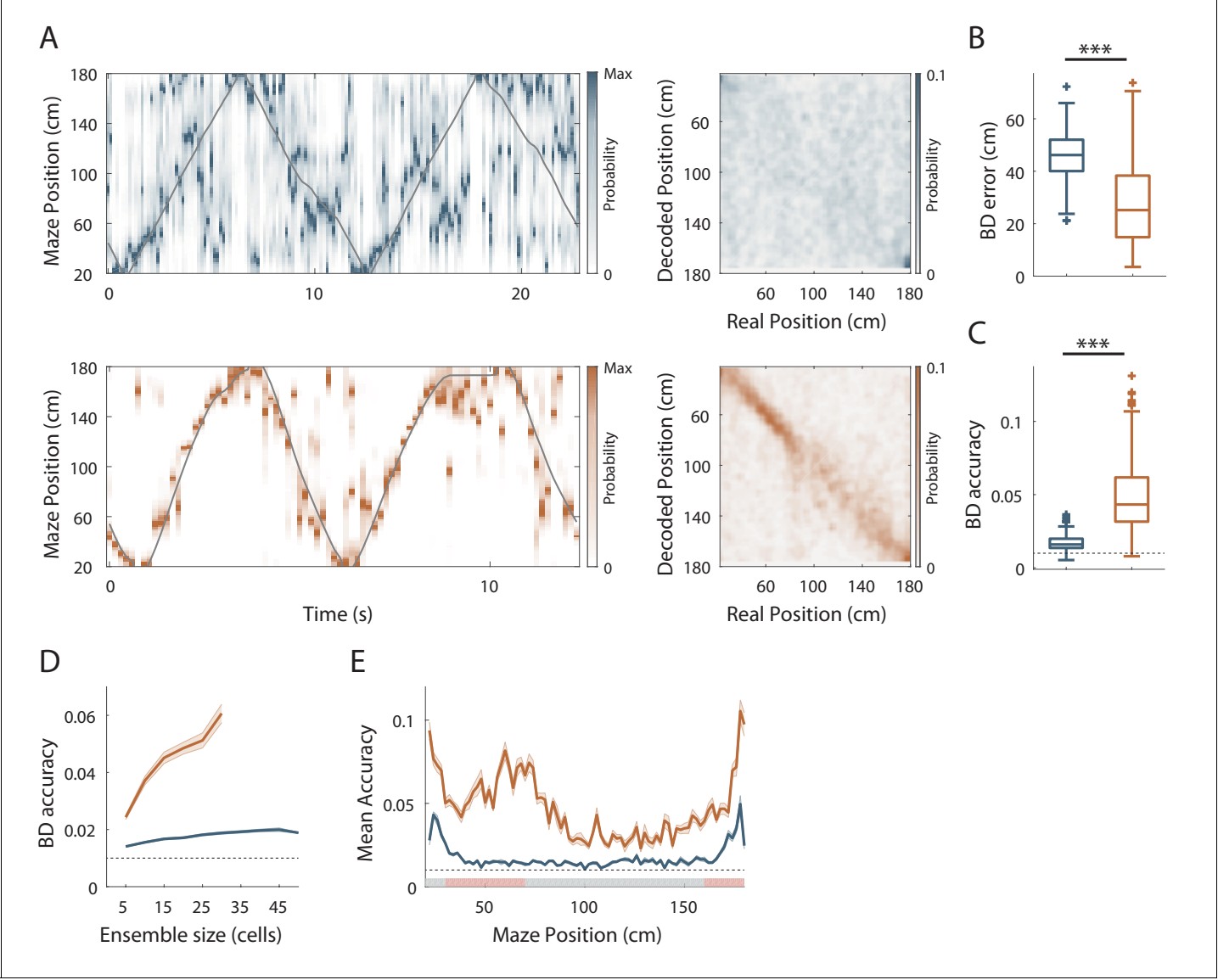

**Figure 3.** Effect of visual cues on spatial coding resolution at the population level. (**A**) Left: Color-coded distribution of the animal position's probability in the virtual track (the reward zones are excluded) computed using a Bayesian decoder (see Materials and methods) at each time window (500 ms) illustrated during four trials in the maze without (top) and with (bottom) objects. Spike trains of active cells were used to compute the animal position's probability. For visualization purpose, position probability is normalized by its maximum at each time bin. The real position is indicated with a solid grey line. Right: Confusion matrix between the real (x-axis) and the decoded position (y-axis) for all recording sessions performed on the track without objects (top) or with objects (bottom). (**B**) Box plots depicting the Bayesian decoding error (BD error) in the maze with and without objects. The BD error was significantly higher in the maze deprived of objects. (**C**) Box plots depicting the Bayesian decoding accuracy (BD accuracy) in the maze with and without objects. The BD accuracy was significantly higher in the maze with objects. (**D**) Mean BD accuracy (solid lines)±SEM (shaded bands) as a function of a subset of active cells in the maze with and without objects. (**E**) Mean BD accuracy (solid lines)±SEM (shaded bands) at each position in the maze with and without objects. The track was divided in two zones: Objects Zone (OZ, in red on the x axis) around the objects and No Object Zone (ØZ, in grey on the x axis) deprived of objects. Note that the decoding accuracy was specifically improved in OZ in comparison to ØZ in the maze with objects.

DOI: https://doi.org/10.7554/eLife.44487.009

The following source data and figure supplements are available for figure 3:

**Source data 1.** Source data for *Figure 3*.
DOI: https://doi.org/10.7554/eLife.44487.012

**Figure supplement 1.** Virtual 3D objects improve hippocampal population coding accuracy across different recording sessions Box plots of the Bayesian decoding error.

*Figure 3 continued on next page*

*Figure 3 continued*

DOI: https://doi.org/10.7554/eLife.44487.010

**Figure supplement 2.** Firing Rate vector decoding in familiar conditions.

DOI: https://doi.org/10.7554/eLife.44487.011

These effects were independent of the decoding method used because similar results were observed using a Firing Rate Vector (FRV) method (*Figure 3—figure supplement 2*; *Wilson and McNaughton, 1993*; *Middleton and McHugh, 2016*). Correlation values were lower in the empty track (ØT: $0.63 \pm 0.008$, n = 180 trials; OT: $0.74 \pm 6.69 \times 10^{-3}$, n = 249 trials; $Z = -10.27$, $p < 10^{-24}$, two-tailed WRS test) and decoding errors were higher (ØT: $49.12 \pm 0.68$ cm, n = 180 trials; OT: $31.31 \pm 1.09$ cm, n = 249 trials; $Z = 11.21$, $p < 10^{-29}$, two-tailed WRS test). Because Bayesian decoding was performed using a drop cell approach, we could measure decoding accuracy for different sample sizes of active cells (*van der Meer et al., 2010*) (*Figure 3D*). Decoding accuracy was positively correlated with sample size in the track with objects but not in the track without objects (*Figure 3D*). Importantly, decoding accuracy was better in OT even if the sample size of active cells used was three time lower than in ØT (to compensate for the three time lower proportion of place cells in this condition; ØT n = 15 vs OT n = 5, $Z = -2.26$, $p=0.02$, two-tailed WRS test; ØT n = 30 vs OT n = 10, $Z = -2.85$, $p=0.004$, two-tailed WRS test; ØT n = 45 vs OT n = 15, $Z = -2.55$, $p=0.01$, two-tailed WRS test). To see if objects could locally increase spatial decoding accuracy, we compared decoding accuracy between OZ and ØZ. While decoding accuracy was uniformly low in the track without objects (OZ: $0.02 \pm 1.48 \times 10^{-3}$, n = 30 spatial bins of 2 cm; ØZ: $0.02 \pm 8.98 \times 10^{-4}$, n = 50 spatial bins of 2 cm; $Z = -1.64$, $p=0.1$, two-tailed WRS test; *Figure 3E*), it was increased in every part of the track with objects but significantly more in OZ compared to ØZ (OZ: $0.06 \pm 0.003$, n = 30 spatial bins of 2 cm; ØZ: $0.04 \pm 2.3 \times 10^{-3}$, n = 50 spatial bins of 2 cm; $Z = -5.21$, $p < 10^{-6}$, two-tailed WRS test; *Figure 3E*). We concluded that local visual cues can globally and locally improve spatial coding accuracy at the population level.

## Fast dynamics of spatial coding resolution tuning upon objects manipulation

Place cells usually appear instantaneously upon exploration of a new environment in area CA1 (*Wilson and McNaughton, 1993*; *Epsztein et al., 2011*). To see if similar dynamics could be observed for the effects of virtual objects on spatial resolution, we manipulated objects online while recording the same ensemble of cells in area CA1. For mice trained in an empty track, we instantaneously added the three objects (which were thus new to the mice) after 20 back and forth trials. Conversely, for mice trained in the track with objects we instantaneously removed the three objects. Objects manipulation had no effect on the proportion of active cells (*Figure 4B*) but a strong impact on the proportion of place cells (*Figure 4A and C*). For mice trained in an empty track, adding objects instantaneously increased the proportion of place cells (from $21.6 \pm 5.3\%$ to $75.0 \pm 4.1\%$; n = 5 sessions in three mice; $t_4 = -35.8$, $p < 10^{-5}$, two-tailed paired *t*-test; *Figure 4A and C*). Thus, a large proportion of cells initially silent or active but nonspatially modulated in the familiar empty track became spatially modulated (40.3%). Most of these cells had on-track fields (81.3%; *Figure 4H*). A majority of cells initially spatially modulated remained place cells (75.7%), while the others became nonspatially modulated or silent. Adding objects also increased place cells' stability ($Z = -4.68$, $p < 10^{-5}$, two-tailed WRS test; *Figure 4E*) and spatial information ($Z = -3.20$, $p=0.0014$, two-tailed WRS test; *Figure 4G*). Local stability was significantly higher in OZ when objects were added (OZ: $0.56 \pm 0.02$, n = 60 bins of 2 cm, 30 in each direction; ØZ: $0.25 \pm 0.02$, n = 100 bins of 2 cm, 50 in each direction; $Z = -8.57$, $p < 10^{-16}$, two-tailed WRS test; *Figure 4I*) but not before ($Z = 1.25$, $p=0.21$, two-tailed WRS test). Place fields' spatial dispersion and out/in field firing ratio were decreased ($Z = 3.55$, $p=0.0004$ and $Z = 1.87$, $p=0.06$, respectively, two-tailed WRS test; *Figure 4D and F*).

On the other hand, removing objects decreased the proportion of place cells (from $71.1 \pm 5.54\%$ to $34.9 \pm 10.9\%$, n = 8 sessions in three mice; $t_5 = 5.54$, $p=0.001$, two-tailed paired *t*-test; *Figure 4A and C*). The spatial information and stability were decreased by this manipulation ($Z = 2.27$, $p=0.02$ and $Z = 4.51$, $p < 10^{-5}$, respectively, two-tailed WRS test; *Figure 4E and G*), while place field out/in

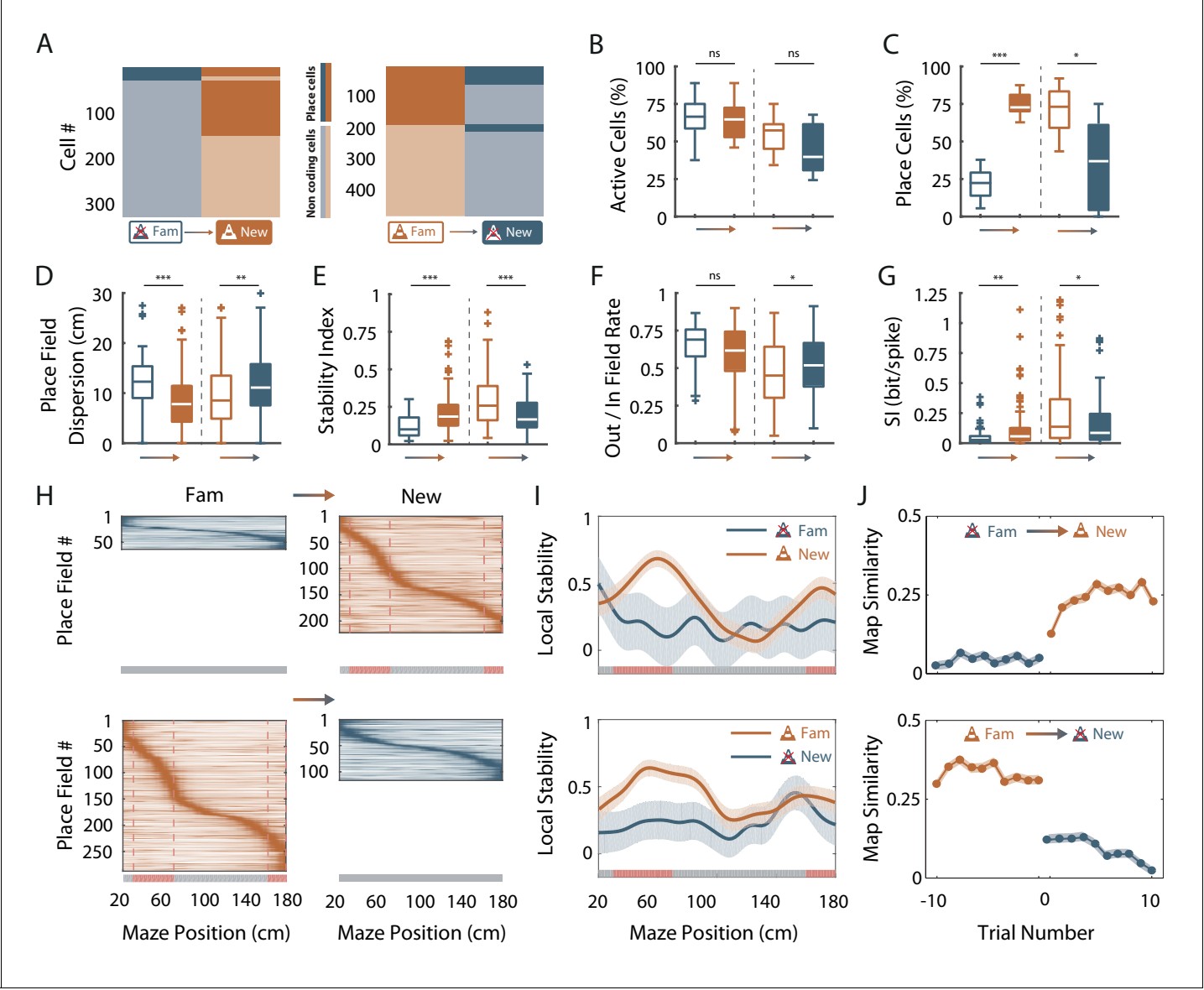

**Figure 4.** Fast dynamics of spatial coding resolution tuning upon objects manipulation. (A) Mosaic plots representing the cells classified as place cells (darker orange and blue) or non-coding cells (i.e. silent or active non-coding, lighter orange and blue) in the familiar and the new mazes. (B–G) Box plots comparing familiar (empty box) and new mazes (filled box) conditions. Two pairs of box plots are illustrated; Left: comparison between the familiar maze without objects (blue, $\varnothing T_{fam}$) and the new maze with objects (orange, $OT_{new}$). Right: comparison between the familiar maze with objects (orange, $OT_{fam}$) and the new maze without objects (blue, $\varnothing T_{new}$). A gradient color arrow shows the way of the transition. Plots show the percentage of active cells (B), the percentage of place cells (C), the Out/In field firing rate (D), the spatial information (SI; E) and the stability index (G). (H) Color-coded mean firing rate maps of place fields recorded in the familiar and new mazes. The color codes for the intensity of the firing rate normalized by the peak rate. The place fields are ordered according to the position of their peak rate in each track (the reward zones are excluded). The tracks were divided into Objects Zones (OZ, in red on the x-axis) around the objects and No Object Zones (ØZ, in grey on the x-axis) deprived of objects. Red dotted lines depict the boundaries of the OZ in the track with objects. (I) Mean local stability index (solid orange or blue lines)±SEM (blue or orange shaded areas) at each spatial bin in the familiar and new mazes (top: from $\varnothing T_{fam}$ to $OT_{new}$; bottom: from $OT_{fam}$ to $\varnothing T_{new}$). (J) Map similarity (see Materials and methods) for 10 trials before and 10 trials after the experimental manipulation (indicated by 0) for $\varnothing T_{fam}$ to $OT_{new}$ (top) and for $OT_{fam}$ to $\varnothing T_{new}$ condition (bottom).

DOI: https://doi.org/10.7554/eLife.44487.013

The following source data and figure supplements are available for figure 4:

**Source data 1.** Source data for *Figure 4*.
DOI: https://doi.org/10.7554/eLife.44487.016

*Figure 4 continued on next page*

*Figure 4 continued*

**Figure supplement 1.** Spatial coding resolution adaptation upon objects manipulations is already visible during the first session in the new condition Box plots comparing familiar (empty boxes, all recording sessions) and new (filled boxes, first recording session) conditions upon objects manipulation.

DOI: https://doi.org/10.7554/eLife.44487.014

**Figure supplement 2.** Virtual 3D objects modulation of hippocampal population coding accuracy upon objects manipulation.

DOI: https://doi.org/10.7554/eLife.44487.015

field firing ratio and dispersion were increased ($Z = -2.01$, p=0.04 and $Z = -3.06$, p=0.002, respectively, two-tailed WRS test; *Figure 4D and F*). After object removal, local stability was not significantly higher in OZ (OZ: 0.24 ± 0.02, n = 60 bins of 2 cm, 30 in each direction) compared to ØZ (0.24 ± 0.02, n = 100 bins of 2 cm, 50 in each direction; $Z = 0.19$, p=0.85, two-tailed WRS test; *Figure 4I*). Importantly, these effects were already observed during the first recording sessions following objects manipulation (*Figure 4—figure supplement 1*). Furthermore, objects manipulations were associated with significant changes of spatial coding resolution at the population level (*Figure 4—figure supplement 2*). We conclude that the effects of local visual cues on place cells' coding observed between familiar tracks can be reproduced with instantaneous objects manipulation.

We next investigated the dynamic of these changes by first calculating the correlation of the firing rate maps of each back and forth trial with the corresponding average firing rate map in the condition with objects (the most stable condition) for 10 trials before (t-1 to t-10) and 10 trials after (t + 1 to t + 10) the manipulation (*Figure 4J*) then comparing the correlation values before and after the manipulation. When objects were added in the empty track, map similarity was significantly higher for the second trial in the new condition (t-1 vs t + 2, n = 598 and n = 608 pyramidal cells, respectively; n = 5 sessions in three mice; $Z = 7.18$, $p<10^{-9}$; Kruskall-Wallis one-way test with post-hoc Bonferroni correction) and then stayed higher from this second trial on (t + 2 vs t + 3, n = 608 and n = 612 pyramidal cells, respectively; n = 5 sessions in three mice; $Z = 1.10$, p=1, Kruskall-Wallis one-way test with post-hoc Bonferroni correction). Conversely, when objects were removed from the familiar track with objects, map similarity dropped already for the first trial in the new condition (t-1 vs t + 1, n = 744 and n = 743 pyramidal cells, respectively; n = 8 sessions in three mice; $Z = 8.80$, $p<10^{-15}$, Kruskall-Wallis one-way test with post-hoc Bonferroni correction) and stayed lower from this first trial on (t + 1 vs t + 2, n = 743 and n = 720 pyramidal cells, respectively; $Z = 0.17$, p=1, Kruskall-Wallis one-way test with post-hoc Bonferroni correction). Thus, the hippocampus can rapidly adapt its spatial coding resolution to local visual cues available in the environment.

## Low proportion of object-responsive cells in OT

Newly activated place cells in OT could correspond to object responsive (OR) cells, which have been recorded in the hippocampus of freely moving rats (*Deshmukh and Knierim, 2013*). These cells tend to discharge systematically near several objects present in the environment. To test this hypothesis, we specifically looked for OR cells in our recordings. For this analysis, we took advantage of the fact that our animals were passing near the same objects in both back and forth trials. Indeed, OR cells should systematically discharge near several objects (if they do not code for objects identity) or the same object (if they in addition code for objects identity) in both back and forth trials. We defined object zones for each individual object (IOZ). Place cells were classified as OR cells if they were bidirectional (firing in both back and forth trials) and had at least one place field in a IOZ corresponding to the same object for both back and forth trials or several place fields in several IOZs corresponding to the same objects in both back and forth trials. In the track without objects no OR cell was detected. In the track with objects, OR cells represented only 2.07% of all place cells. We conclude that the vast majority of newly activated place cells in the presence of objects does not correspond to OR cells.

## Effects of 2D wall patterns on hippocampal spatial coding resolution

We next wondered whether the effect of objects on hippocampal spatial coding resolution could be recapitulated by having more 2D local visual cues in different positions along the track. We thus assessed hippocampal spatial coding in another environment devoid of the original 3D objects but enriched with different wall patterns along the track (Pattern No Objects track or PØT; *Figure 5A*). The percentage of active cells was not affected by the presence of different patterns along the track

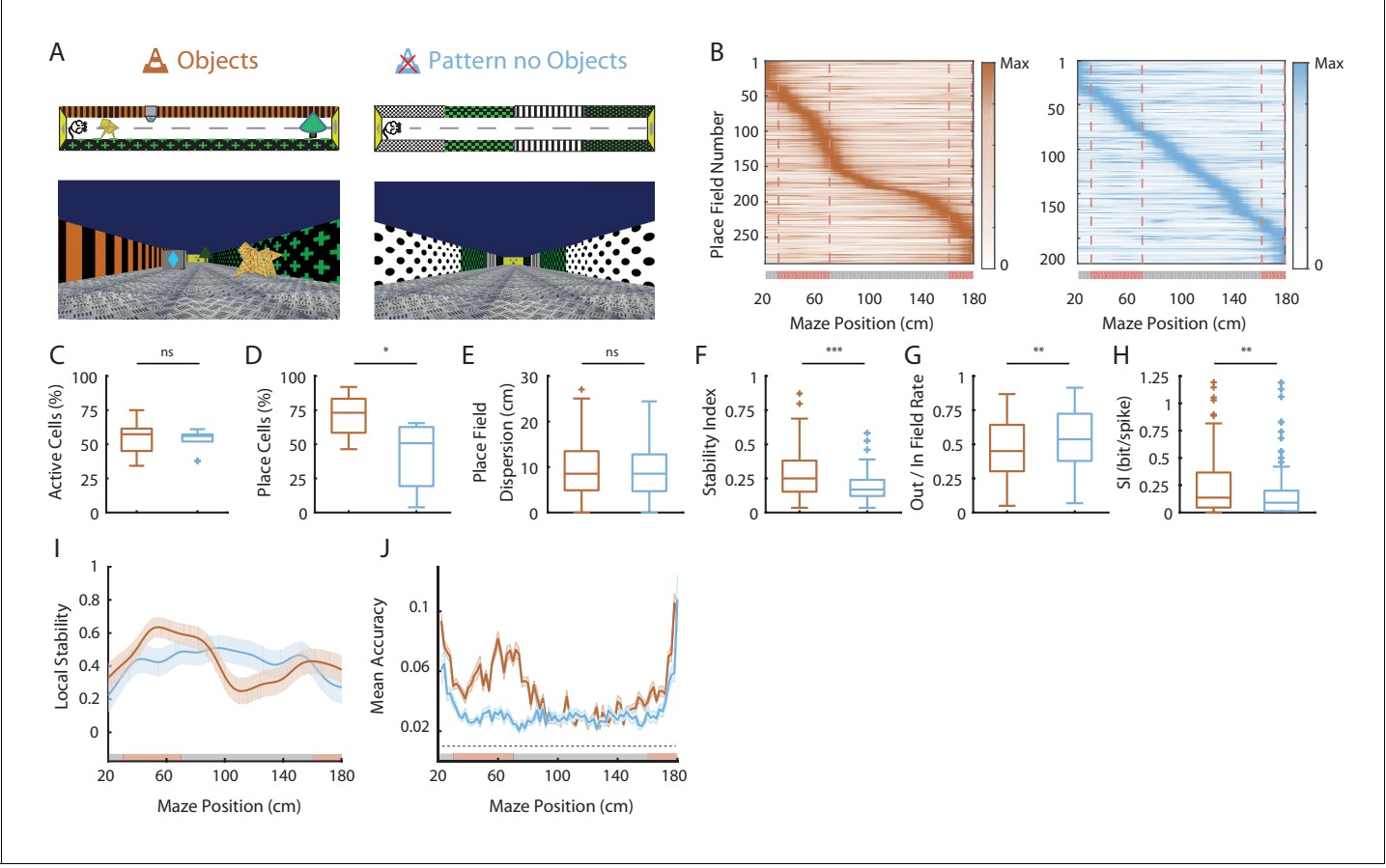

**Figure 5.** Effects of 2D wall patterns on hippocampal spatial coding resolution. (**A**) Schema (top) and picture (bottom) representing the original maze with objects (orange, left) and a maze with patterns on the walls but no objects (PØT; ligth blue, right). (**B**) Color-coded mean firing rate maps for all place fields recorded in the original maze with objects (orange, left) and on the PØT maze (light blue, right). The color codes for the intensity of the firing rate normalized by the peak rate. The place fields are ordered according to the position of their peak rate in each track (the reward zones are excluded). The tracks were divided into Objects Zones (OZ, in red on the x-axis) around the objects and No Object Zones (ØZ, in grey on the x-axis) deprived of objects. Red dotted lines depicts the boundaries of the OZ. (**C–H**) Box plots representing in the original (orange) and pattern no object (light blue) mazes the percentage of active cells (**C**), the percentage of place cells (**D**), the place field dispersion (**E**), the stability index (**F**), the out/in field rate (**G**) and the spatial information (SI; **H**). (**I**) Mean local stability index (solid orange or light blue lines)±SEM (orange or light blue shaded bands) at each position's bin in the original (orange) and pattern no object (light blue) mazes. (**J**) Mean BD accuracy (solid lines)±SEM (shaded bands) at each spatial bin in the original maze with objects (orange) or in the pattern no object maze (light blue).

DOI: https://doi.org/10.7554/eLife.44487.017

The following source data is available for figure 5:

**Source data 1.** Source data for *Figure 5*.
DOI: https://doi.org/10.7554/eLife.44487.018

(OT, 54.64 ± 4.79%, n = 8 sessions in three mice; PØT, 53.63 ± 3.45%, n = 6 sessions in two mice; p=1, one-way Anova test with post-hoc Bonferroni correction; *Figure 5C*). The percentage of place cells among active cells tended to be greater than in ØT (ØT, 18.71 ± 4.33%, n = 7 sessions in three mice; PØT, 42.16 ± 10.58%, n = 5 sessions in two mice; p=0.093, one-way Anova test with post-hoc Bonferroni correction). Also, the percentage of place cells in PØT was significantly lower than in OT (OT, 71.11 ± 5.54%, n = 8 sessions in three mice; p=0.024, one-way Anova test with post-hoc Bonferroni correction; *Figure 5B–D*). Interestingly, place fields were uniformly distributed along the track enriched with patterns (n = 16 spatial bins of 10 cm; p=0.23, test for non-uniformity; *Figure 5B*). This suggests that local 2D visual cues are sufficient to set place fields' position. Place field width was significantly decreased in PØT compared to ØT (ØT, 51.46 ± 3.34 cm, n = 15 place fields; PØT, 41.51 ± 1.17 cm, n = 138 place fields; Z = 2.62, p=0.026, Kruskall-Wallis one-way test with post-hoc

Bonferroni correction). Accordingly, place field dispersion was significantly reduced compared to ØT to a level comparable to OT (ØT, 5.95 ± 0.45 cm, n = 48 place cells; PØT, 4.57 ± 0.21 cm, n = 157 place cells; Z = 2.88, p=0.011, Kruskall-Wallis one-way test with post-hoc Bonferroni correction; *Figure 5E*). Inter-trial firing stability, while significantly higher in PØT compared to ØT (ØT, 0.12 ± 0.01, n = 48 place cells; PØT, 0.19 ± 0.01, n = 157 place cells; Z = 3.72, p=0.0005, Kruskall-Wallis one-way test with post-hoc Bonferroni correction), was significantly lower than in OT (OT, 0.28 ± 0.01, n = 193 place cells; Z = 5.02, p<$10^{-4}$, Kruskall-Wallis one-way test with post-hoc Bonferroni correction; *Figure 5F*). We conclude that local 2D visual cues can improve place fields stability to a certain extend without, however, reaching the level of stability observed in the presence of 3D virtual objects.

We next assessed spatial selectivity and information content in PØT. The ratio of place cells' out-of-field versus in-field firing was lower in PØT compared to ØT (ØT, 0.65 ± 0.02, n = 48 place cells; PØT, 0.53 ± 0.02 cm, n = 157 place cells; Z = 3.38, p=0.002, Kruskall-Wallis one-way test with post-hoc Bonferroni correction) but still significantly higher than in OT (OT, 0.46 ± 0.01, n = 193 place cells; Z = 3.02, p=0.007, Kruskall-Wallis one-way test with post-hoc Bonferroni correction; *Figure 5G*). Accordingly, spatial information content was higher in PØT compared to ØT (ØT, 0.056 ± 0.01, n = 48 place cells; PØT, 0.16 ± 0.02, n = 157 place cells; Z = 4.09, p=0.0001, Kruskall-Wallis one-way test with post-hoc Bonferroni correction) but still significantly lower than in OT (OT, 0.25 ± 0.02, n = 193 place cells; Z = 2.73, p=0.018, Kruskall-Wallis one-way test with post-hoc Bonferroni correction; *Figure 5H*).

Altogether these results indicate that local 2D visual cues can enhance the proportion of place cells among active cells and place cells' coding accuracy but to a lower extend compared to 3D virtual objects.

Finally, local stability in OZ was significantly higher in OT compared to PØT (OT-OZ: 0.52 ± 0.02; PØT-OZ: 0.39 ± 0.016, n = 30 spatial bins of 2 cm for both; Z = 4.85, p<$10^{-5}$ two-tailed WRS test; *Figure 5I*). Accordingly, the same effect could be observed for the decoding accuracy (OT-OZ: 0.07 ± 0.02; PØT-OZ: 0.03 ± 0.01, n = 30 spatial bins of 2 cm for both; Z = −5.83, p<$10^{-8}$ two-tailed WRS test; *Figure 5J*) in agreement with a strong influence of 3D objects on spatial coding resolution.

## Spatial coding resolution in a visually enriched environment

We next wondered whether the hippocampal mapping resolution was maximal in the presence of objects or whether it could be increased by further visually enriching the environment. We thus analyzed hippocampal place cells' coding in another environment containing the original 3D objects but enriched in visual cues such as different wall patterns in different positions along the track and high 3D columns outside the track (EOT, n = three mice; *Figure 6A*). The percentage of active cells was not increased by visually enriching the environment (OT, n = 5 sessions in two mice; EOT, n = 5 sessions in three mice; Z = −0.1, p=1, two-tailed WRS test; *Figure 6C*) nor was the percentage of place cells (OT, n = 5 sessions in two mice; EOT, n = 5 sessions in three mice; $t_8$ = −1.38, p=0.20, two-tailed unpaired t-test; *Figure 6B–D*). However, place fields were uniformly distributed along the track in the visually rich environment (n = 16 spatial bins of 10 cm; p=0.23, test for non-uniformity), thus not clustered around objects as in the visually poor environment (*Figure 6B*). This suggests that local visual cues are important to set place fields' position (*Renaudineau et al., 2007*). However, all other attributes of place fields were not significantly different between the two environments (OT, n = 103 place cells; EOT, n = 132 place cells; out/in field firing ratio: Z = 0.57, p=0.57; Spatial info: Z = 0.42, p=0.67; Dispersion: Z = −1.88, p=0.06; Stability: Z = −0.06, p=0.95; two-tailed WRS test for all; *Figure 6E–H*). When looking at local stability of firing rates, we still observed a significant effect of objects in the visually enriched environment in OZ versus ØZ (OZ, n = 60 spatial bins of 2 cm; ØZ: n = 100 spatial bins of 2 cm; Z = −2.46, p=0.014, two-tailed WRS test; *Figure 6I*). Interestingly, positions near objects were also decoded with a better accuracy using a Bayesian decoder than positions further away in the visually enriched environment (OZ: 0.07 ± 0.004, n = 30 spatial bins of 2 cm; ØZ: 0.057 ± 0.003, n = 50 spatial bins of 2 cm; Z = −4.49, p=0.004, two-tailed WRS test; *Figure 6J*).

Altogether these results suggest that in the presence of local visual cues, hippocampal spatial coding is not further improved by visually enriching the environment. However, place fields locations are influenced by additional visual cues along the track. Interestingly, despite a homogeneous

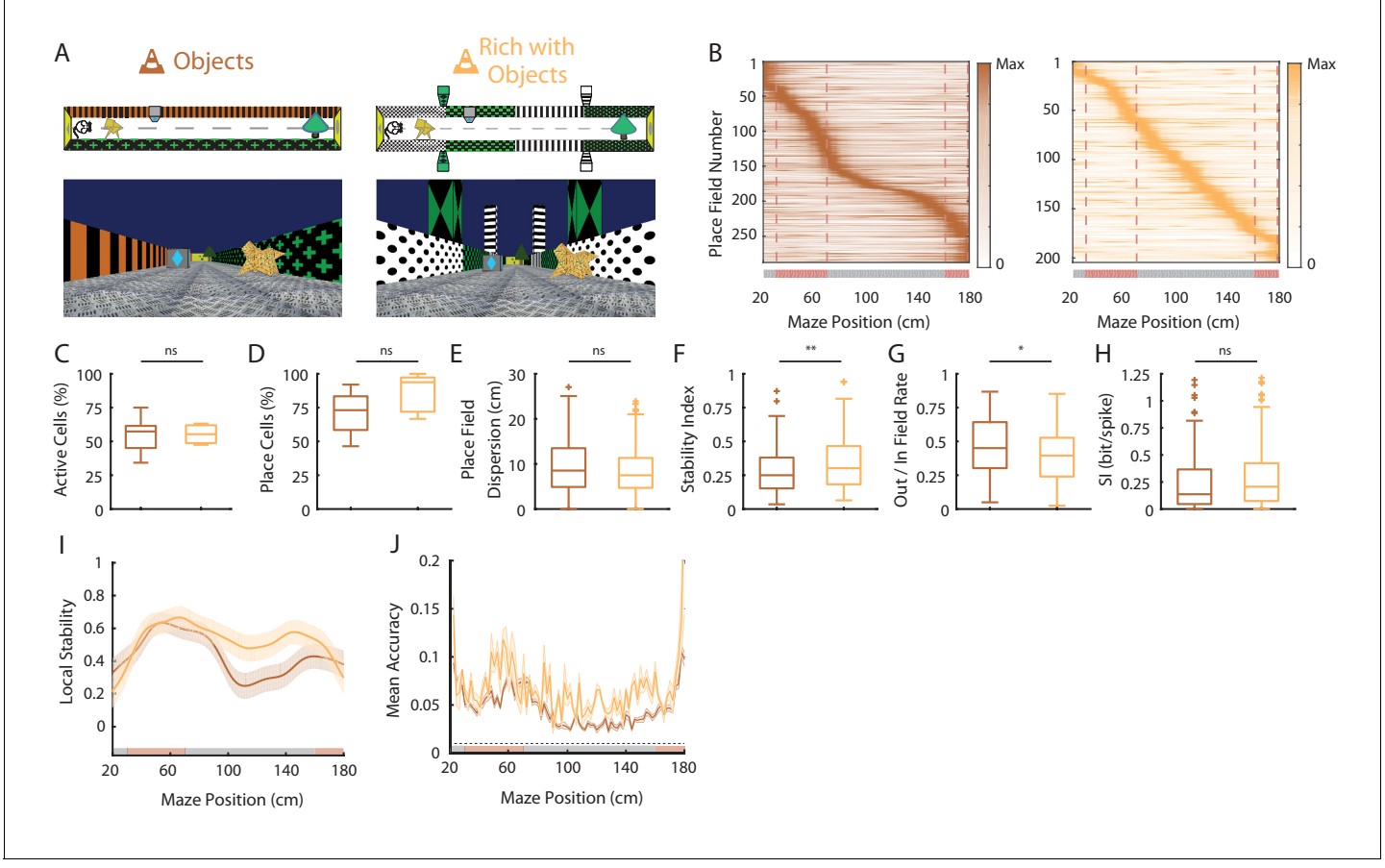

**Figure 6.** Spatial coding resolution in a visually enriched environment . (A) Schema (top) and picture (bottom) representing the original maze with objects (left) and a visually enriched maze with objects (right). (B) Color-coded mean firing rate maps for all place fields recorded in the original maze with objects (orange, left) and on the visually rich maze with objects (yellow, right). The color codes for the intensity of the firing rate normalized by the peak rate. The place fields are ordered according to the position of their peak rate in each track (the reward zones are excluded). The tracks were divided into Objects Zones (OZ, in red on the x-axis) around the objects and No Object Zones (ØZ, in grey on the x-axis) deprived of objects. Red dotted lines depicts the boundaries of the OZ. (C–H) Box plots representing in the original (orange) and pattern no object (light blue) mazes the percentage of active cells (C), the percentage of place cells (D), the place field dispersion (E), the stability index (F), the out/in field rate (G) and the spatial information (SI; H). (I) Mean local stability index (solid orange or yellow lines)±SEM (orange or yellow shaded bands) at each position's bin in the original (orange) and visually rich (yellow) mazes. (J) Mean BD accuracy (solid lines)±SEM (shaded bands) at each spatial bin in the original maze with objects (orange) or in the visually rich maze with objects (yellow).

DOI: https://doi.org/10.7554/eLife.44487.019

The following source data is available for figure 6:

**Source data 1.** Source data for *Figure 6*.
DOI: https://doi.org/10.7554/eLife.44487.020

distribution of place field locations, 3D objects could still locally influence hippocampal population decoding accuracy.

## Effects of local cues on hippocampal temporal coding resolution

The results so far suggest that local visual cues can increase spatial coding resolution when considering the spatial firing rate code. Place cells, however, do not only increase their firing rate inside the place field but also tend to fire at progressively earlier phases of the theta oscillation as an animal moves through the place field (*O'Keefe and Recce, 1993*). This phenomenon, called theta phase precession, is thought to further increase spatial coding resolution because different locations within the place field that are difficult to distinguish based on firing rate alone can be accurately separated

when phase is taken into account. In the temporal domain, increased spatial resolution would thus correspond to increased slope of the phase versus position relationship for identical field sizes.

We first looked for differences in the theta oscillation recorded in the Local Field Potential (LFP) between the two conditions. The mean frequency of the field theta oscillation was not significantly different when mice were running in the track with or without objects (ØT: 6.79 ± 0.12 Hz, n = 9 sessions in three mice; OT: 6.59 ± 0.33 Hz, n = 8 sessions in two mice; $Z$ = 1.26, p=0.20, two-tailed WRS test) but was lower than that reported for mice navigating in real linear tracks (*Middleton and McHugh, 2016*). The power of theta oscillation (theta index see Materials and methods section) was also not significantly different (ØT: 3.31 ± 0.23, n = 9 sessions in three mice; OT: 3.38 ± 0.16, n = 8 sessions in three mice; $t_{15}$ = 0.26, p=0.79, two-tailed unpaired $t$-test). Theta frequency was not modulated by running speed of the animal in ØT (r = 0.02 ± 0.02, n = 9 sessions in three mice; *Figure 7— figure supplement 1A,C*) as previously observed in virtual linear tracks when only distal cues are present (*Ravassard et al., 2013*). Theta frequency-speed modulation was, however, significant in OT (r = 0.08 ± 0.03, n = 8 sessions in three mice; $t_{15}$ = -1.44, p=0.17, two-tailed unpaired $t$-test; *Figure 7—figure supplement 1A,C*). Theta amplitude was similarly modulated by running speed in both conditions (ØT: r = 0.07 ± 0.03, n = 9 sessions in three mice; OT: r = 0.03 ± 0.02, n = 8 sessions in three mice; $t_{15}$ = 0.08, p=0.43, two-tailed unpaired $t$-test; *Figure 7—figure supplement 1B,D*). The proportion of active putative pyramidal cells with significant theta modulation was not different between conditions (ØT: 92.24%, n = 361 active cells; OT: 91.97%, n = 299 active cells; $\chi^2$ = 0.01, df = 1, p=0.89, Chi-Square test). The coupling of spikes to theta oscillation was also not significantly different between conditions in terms of preferred phase (ØT: 203.91˚±2.6, n = 361 active cells; OT: 191.17˚±2.87, n = 299 active cells; p=0.07, circular Kruskal-Wallis; *Figure 7—figure supplement 2A, B*) and strength (mean resultant vector length ØT: 0.18 ± 0.006, n = 361 active cells; OT: 0.19 ± 0.009, n = 155 active cells; $Z$ = −1.63, p=0.1, two-tailed WRS test; *Figure 7—figure supplement 2A,C*).

We then analyzed place cells' theta phase precession. To compensate for decreased spatial stability in the ØT condition, we took into account only trials with good correlation with the average place fields (Spatially Stable Trials or SST) for place cells recorded in the empty track (*Schlesiger et al., 2015*), but included all trials for place cells recorded in the track with objects. The stability index of SST fields in ØT was slightly but significantly higher than the stability index of all fields in OT (ØT, n = 48 SST fields; OT, n = 310 fields; $Z$ = 3.32, p<10$^{-3}$, two-tailed WRS test). The percentage of fields with significant (p<0.05) and negative correlation between phase and position (i.e. precessing fields) was high in the track with objects (40.22%), comparable to that observed in real linear tracks in mice but low in the empty track (7.46%; $\chi^2$ = 26.57, df = 1, p<10$^{-6}$ compared to OT, Chi-Square test). Accordingly, the correlation between phase and position was significantly different from zero for place cells recorded in the track with objects (r = −0.14 ± 0.018, n = 177 fields; p<10$^{-14}$, one sample sign-test; *Figure 7A and B*) but not for those recorded in the track without objects (r = 0.15 ± 0.024, n = 15 fields; p=0.30, one sample sign-test; *Figure 7A and B*). Moreover, phase precession slopes (calculated on normalized place field sizes) were negative and significantly different from 0 for cells recorded in the track with objects (−2.00 ± 0.17 rad/U, n = 177 fields; p<10$^{-14}$, one sample sign-test; *Figure 7C*) but not in the track without objects (1.82 ± 0.44 rad/U, n = 15 fields; p=0.3, one sample sign-test; *Figure 7C*). Similar results were observed when a waveform-based method (which takes into account the asymmetry of theta waves, *Belluscio et al., 2012*) was used to estimate theta phase (*Figure 7—figure supplement 3*).

In the track without objects, the decrease in phase-position correlation could result from the higher inter-trial spatial dispersion, which could lead to spikes at different theta phases for identical positions. To assess this possibility, we performed phase-precession analysis on single-trial-detected fields and averaged the slopes of individual passes (*Schmidt et al., 2009*). The correlation was still negative and significantly different from 0 in OT (r = −0.13 ± 0.025, n = 208 single-trial fields; $t_{207}$ = −5.75, p<10$^{-8}$, one sample sign-test) but not in ØT (r = 0.042 ± 0.04, n = 41 single-trial fields; $t_{40}$ = 0.92, p=0.35, one sample $t$-test). Similarly, the slope of the regression line was negative and significantly different from 0 in OT (−1.27 ± 0.75 rad/U, n = 208 single-trial fields; p<10$^{-3}$, sign-test) but not in ØT (0.74 ± 0.96, n = 41 single-trial fields; p=0.93, sign-test).

Because a low percentage of active cells were place cells in the track without objects, we ran an additional analysis that is independent of place field detection. It exploits the fact that phase precessing cells emit theta paced spikes at a frequency slightly faster than the concurrent LFP theta

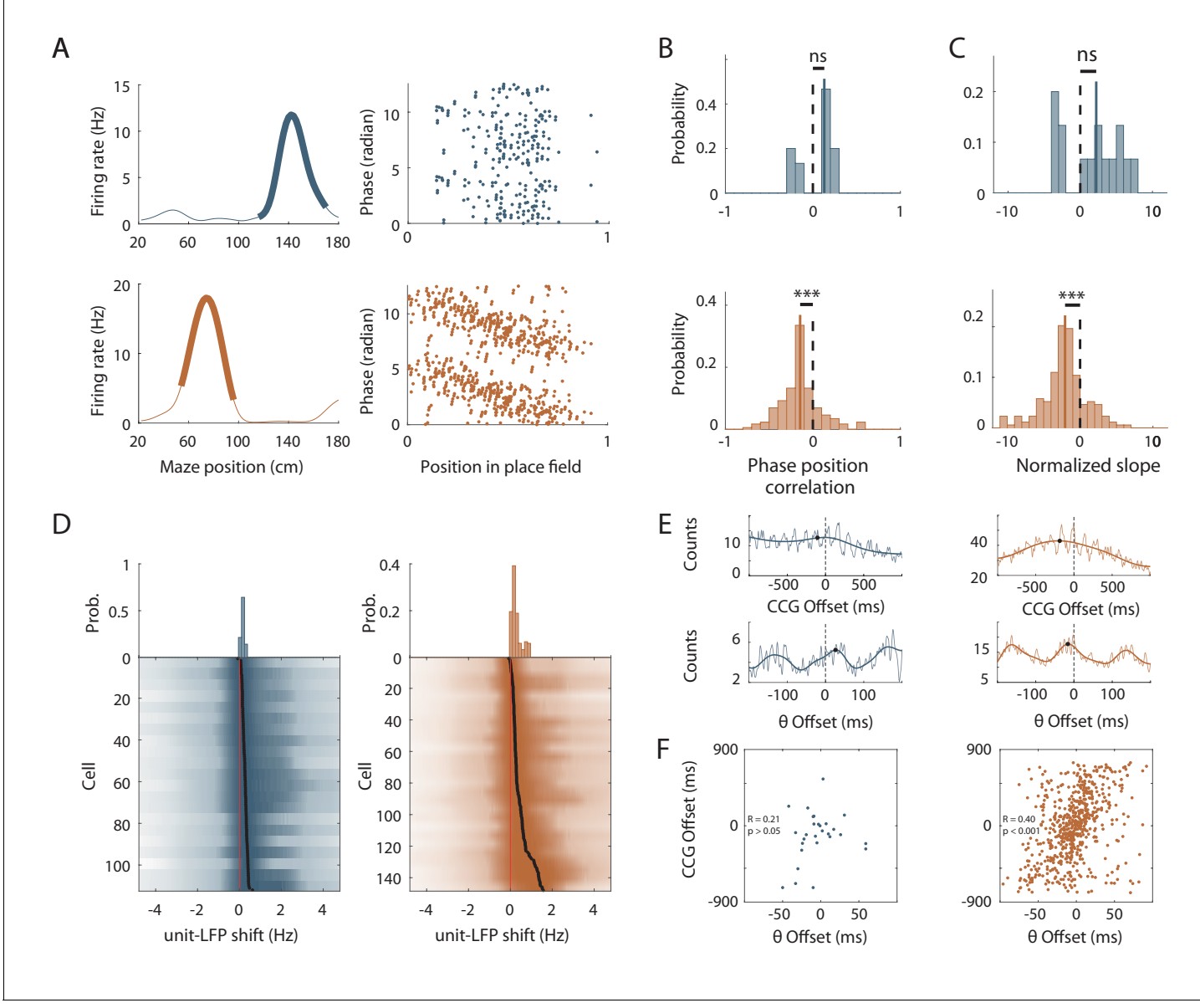

**Figure 7.** Effects of local cues on hippocampal temporal coding resolution . (**A**) Left: Mean firing rate maps of representative CA1 place cells with place fields highlighted by a bold line (left) recorded in the maze without objects (top, only spatially stable trials see Materials and methods section) and with objects (bottom). Right: spikes phase (radian) versus position in the corresponding place fields. (**B–C**) Distribution of significant phase position correlation (**B**) and slopes (**C**) in the condition without objects (top; correlation) and with objects (bottom). The median of the distribution is indicated by a bold line and 0 by a dotted line. (**D**) Color-coded cross-correlogram between the power spectra of neuronal spikes and LFP for each theta-modulated cell recorded on the maze without (bottom left, blue) and with (bottom right, orange) objects. Black dots indicate the maximum of each cross-correlation. Each cross-correlation is normalized by its maximum. Top: Distribution of the maximum cross-correlations used to quantify the frequency shift for all the cells. (**E**) Examples of cross-correlograms computed for two pairs of place cells with overlapping place fields at the behavioral (top) or theta time scale (bottom, see Materials and methods) in order to quantify Cross-Correlogram (CCG) and theta Offsets respectively in no object (blue; left) or object (orange; right) conditions. (**F**) Relationship between 'CCG' and 'theta' offsets in the cross-correlograms of all the spikes in overlapping place fields of neuron pairs recorded in no object (top; blue) and object condition (bottom; orange).
DOI: https://doi.org/10.7554/eLife.44487.021

The following source data and figure supplements are available for figure 7:

**Source data 1.** Source data for *Figure 7*.
DOI: https://doi.org/10.7554/eLife.44487.025
**Figure supplement 1.** Speed modulation of LFP theta frequency and amplitude in OT and ØT.
DOI: https://doi.org/10.7554/eLife.44487.022
*Figure 7 continued on next page*

*Figure 7 continued*

**Figure supplement 2.** Theta modulation of spikes in OT and ØT.
DOI: https://doi.org/10.7554/eLife.44487.023
**Figure supplement 3.** Effect of objects on theta phase precession estimated with a waveform-based approach Distribution of significant phase position correlation (left) and slopes (right) in the condition without objects (top, blue) and with (bottom, orange) when theta phase was detected using a waveform-based approach which takes into account theta waves asymmetry.
DOI: https://doi.org/10.7554/eLife.44487.024

oscillation (*O'Keefe and Recce, 1993*). We performed cross-correlation between the power spectra of neuronal spikes and LFP for all active cells with significant theta modulation of spiking activity (ØT: 112/342 cells = 32.74%; OT: 148/271 cells = 54.6%; $\chi^2$ = 29.59, df = 1, p<$10^{-7}$, Chi-square test) and compared the frequency shift (>0) between spiking and LFP theta oscillations between the two conditions (*Geisler et al., 2007*) (*Figure 7D*). The shift was significantly higher in the OT (0.45 ± 0.03 Hz, n = 148 active cells; *Figure 7D*) versus ØT (0.26 ± 0.01 Hz, n = 112 active cells; Z = −2.74, p=0.006, two-tailed WRS test; *Figure 7D*). Altogether, these results suggest that local visual cues are important for proper theta phase precession in the hippocampus.

To further investigate the effect of local visual cues on temporal coding, we next focused on theta-timescale spike coordination. Previous studies have reported, for place cells with overlapping place fields, a positive correlation between the physical distance separating their place fields' centers and the time (or phase) lag of their spikes within individual theta cycles (*Skaggs et al., 1996*; *Dragoi and Buzsáki, 2006*). Our analysis revealed a strong correlation between theta phase and physical distance in the presence of virtual 3D objects (OT: R = 0.39, n = 629 pairs in three mice; p<$10^{-24}$, Pearson correlation; *Figure 7E,F*) but not otherwise (ØT: R = 0.21, n = 28 pairs in three mice; p=0.26, Pearson correlation; *Figure 7E,F*). These results show that local visual cues are important for temporal coding in the hippocampus beyond theta phase precession.

## Discussion

Our study aimed at determining whether hippocampal spatial coding resolution can rapidly adapt to local features of the environment. We found that spatial coding resolution was increased in the presence of local visual cues through an increase in the proportion of spatially selective place cells among active cells but also enhanced place fields' spatial selectivity and stability. These effects were most prominent in the vicinity of local cues and dynamic upon their manipulations. Local sensory cues also proved to be important for temporal place cell coding such as theta phase precession and theta timescale spike coordination.

Spatial resolution can be improved by pooling information across neurons (*Wilson and McNaughton, 1993*). We found that local visual cues could dramatically increase the number of place cells among active cells (by a threefold factor). The mechanisms of place cell activation are not fully understood. Using sensory-based models of place cells activation (*Hartley et al., 2000*; *Strösslin et al., 2005*; *Barry et al., 2006*; *Sheynikhovich et al., 2009*) one can predict that an increase in the quantity/quality of sensory cues in an environment will enhance the number of place cells coding that environment (*Geva-Sagiv et al., 2015*). However, previous studies using local enrichment with multimodal sensory cues or real objects reported only weak or no effects on dorsal hippocampal cell activity. One study recording in rats navigating between cue-rich and cue-poor parts of the same track reported no effect on the proportion of place cells or on the density of place fields. Furthermore, population vector analysis did not reveal a better disambiguation of nearby locations in the cue-rich part of the track compared to the cue-poor suggesting similar spatial coding resolution (*Battaglia et al., 2004*). Other studies found no overall increase of place cells proportion in 2D environment containing real objects nor a specific bias for place cells to fire near the objects (*Renaudineau et al., 2007*; *Deshmukh and Knierim, 2013*). One possibility to explain the lack of recruitment of additional cells in these studies could be a high recruitment rate of the dorsal hippocampus even in the 'cue poor' condition due to the presence of uncontrolled local cues (*Ravassard et al., 2013*).

We found that place field density was specifically increased near objects. However, studies so far have revealed an homogeneous allocation of place fields in space (*Muller et al., 1987*; *Rich et al.,*

*2014*) in a given environment. Locally activated place cells could correspond to object-responsive (OR) cells, which tend to discharge near several objects or landmarks in a given environment (*Deshmukh and Knierim, 2011*). However, the proportion of these cells is generally low in the dorsal hippocampus (between 6 and 8%) which is in line with the fact that places near real objects are not overrepresented at the population level. We note, however, that this proportion may be underestimated and vary depending on the recording location along both the proximo-distal axis and radial axis (*Geiller et al., 2017*). For example, the distal part of CA1, closer to the subiculum, is more heavily innervated by the lateral entorhinal cortex where OR cells were first discovered (*Deshmukh and Knierim, 2011*) and which is believed to feed information about the 'what' visual stream to the hippocampus (*Knierim et al., 2014*). Extracellular recordings specifically targeting this area in the intermediate hippocampus reported an increased proportion of place cells with smaller place fields in the presence of objects (*Burke et al., 2011*). Interestingly, in this study, the decreased place field size was compensated for by increased place fields' number such that the probability of place cell activation for any point in space was similarly low between the objects and non-objects conditions. However, because objects were distributed all along the maze in this study the local effect of objects was not evaluated. In the present work, the strong increase in the number of place fields in OZ resulted in a significant increase in the proportion of place cells active at these locations despite a local reduction in place field size (OZ: $6.56 \pm 0.32\%/10$ cm, n = 12 spatial bins of 10 cm, six in each direction; ØZ: $4.50 \pm 0.29\%/10$ cm, n = 20 spatial bins of 10 cm, 10 in each direction; $t_{30} = -4.54$, $p<10^{-4}$, two-tailed unpaired t-test). This result shows that while there might be a general mechanism to maintain a constant and low proportion of place cells activated at each position notably between dorsal and ventral parts of the hippocampus (*Skaggs and McNaughton, 1992*; *Maurer et al., 2006*) or between objects and non-objects conditions (when objects are distributed all along the track, *Burke et al., 2011*), spatial coding resolution can nevertheless be increased locally around virtual 3D objects. Whether virtual objects in our study are perceived by mice as real objects is unclear. They notably lack the multisensory component inherent to real objects (*Connor and Knierim, 2017*). Nevertheless, they triggered a large (50%) increase in place cell's proportion which is not compatible with the modest proportion of OR cells reported in our and previous studies.

Instead, our results are more compatible with the hippocampal mapping system using local visual cues to improve its spatial coding resolution. Consistent with this hypothesis, spatial coding was not only quantitatively but also qualitatively increased with a higher spatial selectivity, spatial information content and stability of place fields. Previous studies have reported overrepresentations near rewarded locations (*O'Keefe and Conway, 1978*; *Hollup et al., 2001*; *Dupret et al., 2010*; *Danielson et al., 2016*; *Gauthier and Tank, 2018*; *Sato et al., 2018*) or specific sensory cues (*Wiener et al., 1989*; *Hetherington and Shapiro, 1997*; *Sato et al., 2018*). Importantly, we could also observe overrepresentations of the ends of the maze in ØT, where rewards are delivered and which are associated with prominent visual cues. Nevertheless, End-track fields had a low spatial information content and stability when compared to fields recorded in OT (but similar to On-track fields recorded in the same maze). This argues against increased spatial coding resolution at these locations and further suggests a possible dissociation between overrepresentation and increased spatial coding resolution. Finally, improved coding resolution near objects could be instantaneously tuned upon object manipulation while overrepresentations of specific sensory stimuli or rewarded locations usually takes several days to develop (*Le Merre et al., 2018*; *Sato et al., 2018*).

A previous study in rats specifically compared place cell coding in real and virtual reality environments with distal visual cues only (*Ravassard et al., 2013*). They reported a lower number of spatially modulated cells and lower spatial selectivity in the virtual environment and concluded that distal visual cues alone are not sufficient to fully engage the hippocampal mapping system. Our results complement this study by showing that local visual cues, on the other hand, can increase the proportion of spatially modulated cells (i.e. place cells) among active cells and spatial selectivity. Several factors could explain the specific effect of local visual cues on spatial coding observed in the present study. First, objects could constitute a stable reference point in space to refine estimation of the current subject's position possibly through anchoring of the path integrator system (*McNaughton et al., 2006*; *Poucet et al., 2015*). Close to the objects, this effect could be further reinforced through motion parallax effect. Second, objects as local visual cues have a higher sensory resolution compared to distal visual cues. This can lead to increased spatial coding resolution

according to sensory based models of place cell activation (*Hartley et al., 2000*; *Strösslin et al., 2005*; *Barry et al., 2006*). In line with this, animals tend to increase their sensory sampling rate in order to get a better sensory resolution near important locations (*Geva-Sagiv et al., 2015*). Third, objects, as salient cues in the environment, could modify the attentional state of the animal and favor spatial awareness. Such rise in attention has been shown to increase spatial selectivity in mice (*Kentros et al., 2004*). However, we note that animals were not required to pay close attention to objects locations to perform the task, as task performance was not different between the ØT and OT conditions. Alternatively, objects could represent a source of additional noise in the system thus requiring a higher number of spatially modulated cells and increased spatial selectivity for efficient position coding. However, position decoding was very poor in the maze without objects, which argues against this possibility.

The effects of local cues on spatial coding accuracy were even more pronounced in the temporal domain. Indeed, in the absence of local cues theta phase precession was strongly reduced as observed in rat running in place in a wheel (*Hirase et al., 1999*) despite the presence of place fields and patterns on the walls providing optic flow. When local cues were included, however, hippocampal place cells precessed at a rate comparable to that observed in real environments (*Middleton and McHugh, 2016*). An increased slope of theta phase precession in the presence of real objects was reported before (*Burke et al., 2011*) without a significant change in the correlation between phase and position. Because place fields were smaller in the presence of objects, this increase could result from a scaling of theta phase precession rate with place field size (*Huxter et al., 2003*). In our study, we measured theta phase precession on normalized field sizes and also using single trials. We observed a significant and positive correlation between phase and position in the presence of 3D objects, while this correlation was not different from 0 in the absence of local visual cues. This is consistent with improved temporal spatial information coding in the presence of local visual cues.

To ascertain that this effect did not result from changes in place fields' quality, additional analysis, independent of place fields' detection, were performed (*Geisler et al., 2007*). These analyses also showed that in the presence of local cues individual cells' firing tended to oscillate faster than theta oscillation recorded in the LFP (a sign of theta phase precession) while this was much less the case in the absence of local cues. Importantly, the frequency and power of the theta oscillation recorded in the LFP and the coupling of putative pyramidal cells' firing to this oscillation were also not significantly different between conditions and cannot explain observed differences. The only difference was an attenuation of theta frequency speed modulation in the absence of local cues while theta amplitude vs speed modulation was equivalent in both conditions. A similar absence of theta frequency vs speed modulation (with intact theta amplitude vs speed modulation) was observed in rats navigating virtual reality environments in the absence of local visual cues (*Ravassard et al., 2013*). However, in this study, theta phase precession was unaffected. Thus, the link between an absence of theta frequency vs speed modulation and reduced theta phase precession is not straightforward. Future studies are needed to decipher the mechanisms of the effect of local cues on theta phase precession. Theta phase precession is thought to be involved in the generation of theta sequences, where the time lags between spikes of place cells with overlapping place fields are proportional to the distance separating those fields. This so-called theta sequence compression is thought to be important for spatial memory. Here, we found that theta timescale coordination could be observed in the presence of 3D objects only. This suggests that local sensory cues are important for temporal coding beyond theta phase precession.

Altogether, our results show that enriching an environment with local visual cues allows coding at higher spatial resolution with a high number of spatially modulated cells, smaller firing fields, increased spatial selectivity and stability and good theta phase precession/theta timescale spike coordination. The use of virtual reality raises a growing interest in the field of neuroscience to study spatial cognition in rodents but also in non-human and human primates (*Epstein et al., 2017*). Our results suggest that enriching these environments with local visual cues could help comparing spatial coding in real and virtual environments.

We observed that local visual cues induce a rescaling of spatial coding which is both global and local. What would be the benefit of this rescaling? In the wild, rodents can travel kilometers away from their home to food locations through empty fields (*Taylor, 1978*). Mapping all parts of explored environment at high resolution would require a very large number of neurons and

computational power (*Geva-Sagiv et al., 2015*). Accordingly, place fields tend to be larger in bigger environments (*Fenton et al., 2008*) and the statistics of new place cells recruitment as an environment becomes bigger are non-uniform (*Rich et al., 2014*). Thus, there might be a computational benefit to be able to map at high resolution important places like home base or food locations and to map at lower resolution long transition routes between those locations (*Geva-Sagiv et al., 2015*). Such resolution could depend on the number of local sensory information as presented here. Future work should decipher whether increased spatial coding resolution is associated with better navigational accuracy and spatial memory.

## Materials and methods

### Animals
Data were acquired from 11 male mice C57BL/6J (Janvier/Charles River) between 8 and 12 weeks during the recording phase (weight: 21–23.6 g). The mice were housed 2 or three per cages before the first surgery and then individually with 12 inverted light/dark cycles. Trainings and recordings occurred during the dark phase.

### Ethics
All experiments were approved by the Institut National de la Santé et de la Recherche Médicale (INSERM) animal care and use committee and authorized by the Ministère de l'Education Nationale de l'Enseignement Supérieur et de la Recherche following evaluation by a local ethical committee (agreement number 02048.02), in accordance with the European community council directives (2010/63/UE).

### Surgical procedure to prepare head fixation
A first surgery was performed to implant a fixation bar later used for head-fixation. Animals were anesthetized with isoflurane (3%) before intraperitoneal injection of ketamine (100 mg/Kg) mixed with xylazine (10 mg/Kg) supplemented with a subcutaneous injection of buprenorphine (0.06 mg/Kg). Two jeweller's screws were inserted into the skull above the cerebellum to serve as reference and ground. A dental cement hat was then constructed leaving the skull above the hippocampi free to perform the craniotomies later on. The free skull was covered with a layer of agarose 2% (wt/vol) and sealed with silicon elastomer (Kwik-Cast, World Precision Instruments). A small titanium bar (0.65 g; 12 × 6 mm) was inserted in the hat above the cerebellum to serve as a fixation point for a larger head plate used for head fixation only during trainings and recordings.

### Virtual reality set up
A commercially available virtual reality system (Phenosys Jetball-TFT) was combined with a custom designed 3D printed concave plastic wheel (center diameter: 12.5 cm; side diameter: 7.5 cm; width: 14 cm, covered with silicon-based white coating) to allow 1D movement with a 1/1 coupling between movement of the mouse on the wheel and movement of its avatar in the virtual reality environment. This solution was preferred to the original spherical treadmill running in a X-only mode (which takes into account only rotations of the ball in the X axis to actualize the position of the avatar in the virtual reality environment) which also allows 1D movement but with a more variable coupling between movement of the mouse on the treadmill and its avatar in the virtual reality environment. The wheel was surrounded by six 19-inches TFT monitors, which altogether covered a 270 degrees angle. Monitors were elevated so that the mice's eyes level corresponded to the lower third of the screen height to account for the fact that rodents field of view is biased upward. The head fixation system (Luigs and Neumann) was located behind the animal to not interfere with the display of the virtual reality environment. The virtual reality environment was a virtual 200 cm long and 32 cm wide linear maze with different patterns on the side and end walls and virtual 3D objects (see virtual reality environments section). Movement of the wheel actualized the mouse's avatar position. The mouse could only perform forward or backward movements but could not turn back in the middle of the track (see training section).

## Virtual reality environments

### No objects track (ØT)

Each side wall had a unique pattern (black and orange stripes on one wall; green crosses on black background on the other wall). End-walls had grey triangular or round shapes on a yellow background (*Figure 1A*).

### Objects track (OT)

This maze was identical to the ØT maze concerning wall patterns and dimensions but three virtual objects were included on the sides between the animal trajectory and the walls (*Figure 1A*). The objects were a yellow origami crane (dimensions: 9 × 9 × 7 cm; position: 37 cm from end wall), a blue and grey cube (dimensions: 5 × 5 × 5 cm; position: 64 cm from end wall) and a tree (15 × 15 × 22 cm; position: 175 cm from end-wall). The animal could neither orient toward the objects nor get any sensory feedback from them by any other mean but vision.

### Pattern no objects track (PØT)

This maze had the same dimensions as the previous mazes, but the side walls had distinct symmetrical patterns in different locations along the maze (50 cm long; black dots on white background, black and green squares, black and white stripes and green crosses on black background).

### Enriched objects track (EOT)

This maze was identical to the Pattern No Objects Track (PØT) and included the same virtual reality objects (identical in dimensions and locations) to those of the Objects Track (OT) maze. Outside the maze walls, two large 3D columns were positioned on each side (dimensions 8 × 8×47 cm; positions 58 and 143 cm from end wall) to provide additional visual cues.

## Training

Mice were first habituated to the experimentalist through daily handling sessions of 20 min or more that continued throughout the experiment. After a 3 days post-surgery recovery period, mice were water-deprived (1 ml/day, including the quantity of water taken during the training). After 2–3 days of water deprivation, they were progressively trained to run in the virtual reality set up. First, mice were familiarized with running head-fixed on the wheel for water rewards in a black track (screens always black). During these sessions, animals received as a reward sweetened water (5% sucrose) for each 50 centimeters run on the wheel. When animals were comfortable with the setup, they were trained to run in one of three linear virtual tracks (familiar track) assigned randomly. When animals reached the end of the track, a liquid reward delivery tube extended in front of the animal and animal had to lick to get the reward (a 4 µL drop of water of 5% sucrose). Animals were then teleported in the same position but facing the opposite direction of the maze and had to run up to the end of the maze in the opposite direction to get another reward. Animals were initially trained during 15 min sessions. Session time was progressively increased to reach 60 min. *Ad libidum* water access was restored if the weight of the animal decreased beneath 80% of the pre-surgery weight at any stage during training.

## Recording procedure

When animals reached a stable behavioral performance (at least one reward/minute during 60 min), we performed acute recordings using silicon probes (4/8 shanks; A-32/A-64 Buzsáki Probe, Neuronexus; see *Figure 1—figure supplement 1*). On the day before the first recording session, animals were anesthetized (induction: isoflurane 3%; maintenance: Xylazine/Ketamine 10/100 mg/Kg supplemented with Buprenorphine 0.1 mg/Kg) and a craniotomy was drilled above one hippocampus (centered on a location −2 mm posterior and ±2.1 mm lateral from bregma). The craniotomy was covered with agarose (2% in physiological saline) then sealed with silicon elastomer (Kwik-Cast, World Precision Instruments). This craniotomy was used to record acutely during 2–3 consecutive days (with the probe lowered in a new location every time). Then a second craniotomy was performed over the other hippocampus following the same procedure and recordings were performed during 2–3 additional days. Before each recording session, the backside of the probe's shanks was covered with a thin layer of a cell labeling red-fluorescent dye (DiI, Life technologies) so that its

location (tips of the shanks) could be assessed post-hoc histologically. The silicon probe was then lowered into the brain while the animal was allowed to walk freely on the wheel with the screens displaying a black background. The good positioning of the probe with recording sites in the CA1 pyramidal cell layer was verified by the presence of multiple units showing complex spike bursts on several recordings sites and the recording of sharp-wave ripples during quiet behavior. After positioning of the silicon probe the virtual reality environment was displayed on the screen. On the day of the last recording in each hippocampus, the backside of the probe's shanks was covered with a thin layer of a cell labeling red-fluorescent dye (DiI, Life technologies) so that its location (tips of the shanks) could be assessed histologically post-hoc. All mice (n = 11) experienced a familiar environment (either ØT, OT, PØT or EOT) for around 20 back and forth trials. For mice trained in ØT or OT (n = 3 and 3, respectively), this first exploration was followed, after 3 min of free running with the screens displaying a black background, by exploration of a new environment, identical to the previous one except for the presence of the three 3D objects (objects were added for mice trained in ØT and removed for mice trained in OT) for another 20 consecutive back and forth trials. For some of these mice (n = 2 for ØT, n = 2 for OT, n = 2 for PØT and n = 2 for EOT) sessions in the familiar track and novel track were divided into two sub-sessions interleaved by 3 min of free running with the screens black. The two sub-sessions in the familiar environment and the new environment were pulled together for analysis. Note that animals stayed head-fixed on the wheel surrounded by screens during the entire recording session.

## Data acquisition and pre-processing

The position of the animal in the virtual maze was digitalized by the virtual reality controlling computer (Phenosys) and then sent to a digital-analog card (0–4.5V, National Instrument Board NI USB-6008) connected to the external board (I/O Board, Open Ephys) of a 256 channels acquisition board (Open Ephys). Neurophysiological signals were acquired continuously on a 256-channels recording system (Open Ephys, Intan Technologies, RHD2132 amplifier board with RHD2000 USB interface board) at 25,000 Hz. Spike sorting was performed semi-automatically using KlustaKwik (*Rossant et al., 2016*; https://github.com/klusta-team/klustakwik). Clusters were then manually refined using cluster quality assessment, auto- and cross-correlograms, clusters waveforms and similarity matrix (Klustaviewa, *Rossant et al., 2016*).

## Data analysis

Data analysis was performed in the MATLAB software environment and the source code is available from GitHub (*Marti et al., 2019*; copy archived at https://github.com/elifesciences-publications/codes_bourboulou_marti_2019).

## Reward and object zones definition

The reward zones, located between the maze extremities and 10% of the track length (0–20 cm and 180–200 cm), were not considered in the analysis. The object zone was composed of two zones, one from 30 to 70 cm including both the origami crane and the cube and the other from 160 to 180 cm including the tree.

## Firing rate map

The maze was divided into 100 spatial bins measuring 2 cm. For each trial, the number of spikes and the occupancy time of the animal in each spatial bin were calculated to obtain the spikes number vector and the occupancy time vector, respectively. These vectors were smoothed using a Gaussian filter with a half-width set to 10 spatial bins. Spikes occurring during epochs when velocity was lower than 2 cm/s were removed from all analysis. The smoothed spikes number vector was divided by the smoothed occupancy time vector to obtain the firing rate vector for each trial. The firing rate vectors were pooled for a specific condition (e.g. Familiar Objects Track) and direction of the animal (e.g. back) to generate a firing rate map. These pooled vectors were also averaged to provide the mean firing rate vector, corresponding to the mean firing rate for each spatial bin.

## Pyramidal cell classification

Cells with a mean firing rate lower than 20 Hz and either a burst index (*Royer et al., 2012*) greater than 0 or the spike duration greater than 0.4 ms were classified as putative pyramidal neurons. They were classified as interneurons otherwise. To compute the proportion of active putative pyramidal cells, only sessions with at least 15 recorded neurons were included.

## Active cells classification

A cell was considered as active when the mean firing rate was greater than 0.5 Hz, the peak firing rate was greater than 1.5 Hz and the cell fired at least one spike in 50% of the trials. These three criteria had to be verified in either the forth or back direction.

## Place fields detection

To detect a mean place field, a bootstrap procedure was performed. For each trial, a new spikes train was generated using a Poisson process with λ equal to the mean firing rate of the trial and a 1 ms time interval. A 'randomized' firing rate map was then generated and the mean firing rate vector was determined and compared with the mean firing rate vector from the initial rate map. This operation was repeated 1000 times to determine a p-value vector (p-value for each 2 cm spatial bin). Place fields candidates were defined as a set of more than three continuous spatial bins associated with p-values lower than 0.01. Two place fields were merged when the distance between their closest edges was at most equal to five spatial bins (10 cm). Place fields' edges were extended by at most five spatial bins (for each edge) when the p-value was below 0.30 for these bins. A field with a size greater than 45 spatial bins (90 cm) was not considered as a place field. To validate a mean place field, the cell had to verify a stability criterion. Spatial correlations were calculated between the firing rate vector of each trial and the mean firing rate vector. The spatial bins corresponding to other detected place fields were not considered in the spatial correlations. The place field was validated if the spatial correlations were greater than 0.60 for at least 40% of trials. Unless specified, when several mean place fields were detected, only the place field with the highest peak was conserved. An active cell with at least one place field in one direction was considered as a place cell. To compute the proportion of place cells, only sessions with at least nine active cells were included.

The same procedure was applied to detect place fields per lap without the stability criterion, which cannot be calculated on single trials. A place field per lap was conserved if it overlapped at least one spatial bin with the closest mean place field.

## Stability index

The stability index of a cell was computed as the mean of the spatial correlations between all pairs of firing rate vectors. This way, the cell stability index takes into account the activity patterns from all the trials and provides a reliable quantification of the inter-trial reproducibility of the cells activity. Note that this stability index is different from usual stability indexes based on correlations of mean firing rates between even and odd trials or two halves of the same recording session thus values obtained cannot be directly compared.

## Spatial Information

The spatial information (SI) was calculated according to the following formula (*Skaggs et al., 1996*):

$$SI = \sum_{i=1}^{N} \left[ \frac{FR_i}{FR} \times \frac{OT_i}{OT_T} \times log_2 \left( \frac{FR_i}{FR} \right) \right]$$

where N is the number of spatial bins (N = 100), $FR_i$ is the mean firing rate determined in the i-th spatial bin, $FR$ is the mean firing rate, $OT_i$ is the mean occupancy time determined in the i-th spatial bin, $OT_T$ is the total occupancy time based on the mean occupancy time vector.

As another measure of spatial information, we computed the Mutual Information using the following formula:

$$MI = \sum_{i=1}^{N} \sum_{j=1}^{4} p_{i,j} log_2 \left( \frac{p_{i,j}}{p_i \cdot p_j} \right)$$

where N is the total number of spatial bins, $p_i$ is the occupancy probability of the animal in the i-th spatial bin, $p_j$ is the probability to obtain a firing rate amongst one of four non overlapping quartiles of firing rates and $p_{i,j}$ is the joint probability of the animal to be in the i-th spatial bin with a firing rate in the j-th quartile. The Mutual Information was then normalized with a surrogate-based distribution to correct possible bias due to basal firing rate (*Souza et al., 2018*).

### Out/in-field firing ratio
The out/in-field firing ratio was computed as the ratio between the mean firing rate outside the mean place field (excluding secondary place fields) and the mean firing rate inside the mean place field.

### Place field dispersion
A place field dispersion measure has been computed to quantify how much each place field per lap was dispersed around the mean place field. The place field dispersion (PFD) was calculated according to the following formula:

$$PFD = \frac{L}{N} \left[ \frac{1}{M} \sum_{i=1}^{M} (C - C_i)^2 \right]^{\frac{1}{2}}$$

where C is the center of the mean place field, $C_i$ is the center of the field in the i-th lap and M is the number of laps with a single-trial detected field, L is the total length of the maze and N is the number of spatial bins. The center of a place field was defined as the spatial bin with the highest firing rate.

### Place field width
Place field width was computed as the distance between the place field edges and only determined for entire place fields. A place field was considered as complete when its firing rate increased above 30% of the difference between highest and lowest place field activity and then dropped below this threshold.

### On-track and end-track fields
A mean place field was considered as End-Track field if the peak of the field was located at the beginning of the reward zone (i.e. at the 11-th or the 90-th spatial bin). All other fields were classified as On-Track fields.

### Distribution of place fields' position
To statistically assess whether the place fields were non-uniformly distributed in the maze, we tested the null hypothesis that all fields were uniformly distributed. Based on this hypothesis, the total number of place fields was redistributed with an equal probability to be in each 10 cm spatial bin. The standard deviation of this uniform distribution was then compared to the initial distribution. This operation was repeated 1000 times (bootstrap procedure) to obtain a p-value, corresponding to the probability of the place fields to be uniformly distributed. When this p-value was lower than 0.05, the null hypothesis was rejected and the distribution was considered as non-uniform. To ensure that single values of place fields' percentage in a given bin did not make the distribution non-uniform, values greater than the 93-th percentile and lower than the 6-th percentile have been excluded from the initial distribution.

### Local stability
A local stability index was developed to assess how consistent a firing rate was over the laps for a given spatial bin. To this end, two mean firing rate vectors were calculated, in the neighborhood of each spatial bin (2-spatial bins half-window) for even and odd trials. Local stability index was defined as the spatial correlation between these two vectors for a given spatial bin.

## Position decoding

To address how informative the firing rates of the CA1 pyramidal cells ensemble were about the position of the animal in the different virtual environments, we used Bayesian decoding and Firing Rate Vectors (FRV) methods. For each time window, the distribution of the animal position probability across the whole maze was calculated using the firing activity of all active cells (place cells and non place cells). The mode of this distribution (maximum of probability) was chosen as the decoded position for a given time window. We used a classical 'memoryless' Bayesian decoder (*Brown et al., 1998*; *Zhang et al., 1998*). The decoding of the spikes data was restricted to periods when the animal was running (speed >2 cm/s) or with good Theta/Delta ratio and cross-validated using the 'leave one out' approach. We computed the animal's probability to be in each spatial bin $x$ (2 cm) knowing that N cells fired $n$ spikes in a time window according to the following formula:

$$P(x|n) = C(\tau, n)P(x)\left(\prod_{i=1}^{N} f_i(x)^{n_i}\right) exp\left(-\tau \sum_{i=1}^{N} f_i(x)\right)$$

with $P(x)$ a uniform spatial prior, $f_i(x)$ the average firing rate of the neuron $i$ over $x$ (i.e. the tuning curve over the position), $n_i$ the number of spikes emitted by the neuron $i$ in the current time window and $\tau$ the length of the time window (150 ms; non-overlapping) and $C(\tau, n)$ a normalization factor intended to set the posterior probability for one time window to 1. This formula assumes that the spikes trains obey to a Poisson process and that cells activity is independent. Position decoding was also performed using the FRV method (Middleton and McHugh, 2016). For each 100 ms time bin, the Pearson correlations were calculated between firing rates across all cells and the mean firing rates from all cells for a given spatial bin. A decoding error was defined as the absolute value of the difference between decoded and real position. Accuracy was defined as the probability at the real position in a particular time bin. To ensure that the position decoding was not influenced by the number of cells, a drop cell approach was performed (van der Meer et al., 2010). Briefly, for M recorded active cells, the position was decoded using k different subsets of cells with increasing sizes 5*k with k ranging from 1 to the last multiple of 5 < M. For the k-th subset, the decoding was repeated 50 times using 5*k randomly selected cells and the median value of probabilities for a given time and spatial bin was chosen as the final probability. The presented results were computed for a subset composed of 20 cells (k = 4).

## Map similarity over trials

To analyze the dynamic of the changes of spatial representation between familiar and novel conditions, map similarities were performed for 10 back and forth trials before and after the experimental manipulation. For each active putative pyramidal cell, map similarities consisted of the Pearson correlation between the firing rate map of each back and forth trial and a template firing rate map. This template firing rate map was calculated as the average of the firing rate map from all the laps in the condition with objects (most stable condition). The maps corresponding to back (forth) trials were correlated to the mean back (forth) trial map in the object condition and the correlations values were averaged to obtain a single value for this back and forth trial. When map similarity was determined for a lap in the object condition, the template firing rate map was computed without it.

## Object-responsive cells detection

OR cells tend to discharge systematically at the location of several objects (if they do not code for object identity) present in the environment or at least one object (if they in addition code for object identity). For this analysis, we took advantage of the fact that our animals were passing near the same objects in both back and forth trials. We defined individual objects zones (IOZ), one for each object. For a given object, IOZ corresponded to all spatial bins occupied by the object. Here are the IOZ defined for each object in both directions: origami crane: 30–46 cm, cube: 60–70 cm and tree 164–180 cm. Place cells were classified as OR cells if they were bidirectional (firing in both back and forth trials) and had at least one place field in a IOZ corresponding to the same object for both back and forth trials or several place fields in several IOZs corresponding to the same objects in both back and forth trials.

## Phase precession analysis

Phase precession was calculated on all spikes (above speed threshold) for the track with objects but restrained to Spatially Stable Trials (SST) in the no object condition to equalize stability between both conditions. SST consisted of at least three trials where the in-field correlation with the mean place field exceeded 0.6. To assess theta phase precession, the Local Field Potential (LFP) of the channel with the highest number of pyramidal cells (*Skaggs et al., 1996*) was filtered (4th order Chebyshev filter type II) in the theta band (4–12 Hz). The instantaneous theta phase for each time bin (1 ms) was determined by two different methods: either using the Hilbert transform of the filtered LFP or a waveform-based approach (*Belluscio et al., 2012*). In the later method, cycles extrema were detected in the wide-band signal (1–40 Hz) in each half cycles defined by a zero-crossings of a narrow-band filter (4–10 Hz). LFP theta band phase was then estimated by a linear interpolation between peaks, through and each half cycle in order to preserve theta asymmetry. Both methods produced similar results. Thus, the theta phases used in this paper were obtained using Hilbert transform (unless noted). Only theta phase locked cells were considered in the following analysis (non-uniform phase distribution, p<0.05, Rayleigh test). Circular linear analysis was used to determine the correlation strength and slope value of the relation between spikes phases and normalized positions (0–1) through the mean place field (*Kempter et al., 2012*). Briefly, the phase precession slope was computed with a linear regression model between circular (spike phases) and linear (animal's position) data. The slope of the regression was used to scale the animal's position and to transform it into a circular variable. A circular-circular correlation could thus be computed on the data to assess the strength of the relationship between spike phases and animal's position. A significance value was determined by re-computing the correlation values for 1000 permutations of the spikes position.

Analysis of phase precession on single-trial detected fields was also performed (*Schmidt et al., 2009*). Phase precession slope and correlation values were computed similarly to the previously described method. The single lap slope and correlation values were averaged only for sessions with at least three significantly precessing trials where the cell emitted a minimum of four spikes inside the mean place field.

## Unit-LFP shift and spike phase spectrum

To quantify phase precession independently of the position of the animal and the place field detection, Unit-LFP shift was used. For all active putative pyramidal cells, a discreet multitaper spectrum in the theta band (4–12 Hz) of the cell's spikes was performed (mtpointspectrum, Chronux 2.11; http://chronux.org/)) as well as the continuous multitaper spectrum of the simultaneously recorded LFP (mtspectrumc, Chronux 2.11). A theta modulation index (*Mizuseki et al., 2009*) was defined for each cell spike spectrum as the mean power around the peak theta frequency ±0.5 Hz divided by the mean power below 5 Hz or above 9 Hz. A cell was considered as theta modulated if this index was greater than 1.4. The cross correlogram was then calculated for theta modulated cells to determine the lag in the theta band between the LFP and the cells' spectrum (*Geisler et al., 2007*). A positive lag indicates that the cell is firing faster than the concurrent LFP.

## Speed modulation of theta frequency and amplitude

The instantaneous theta frequency was computed from the instantaneous theta phase extracted from the Hilbert transform of the filtered LFP in the theta band. For each time $t_i$, the instantaneous theta frequency ($F_\theta(t_i)$) was determined based on the unwrapped phase:

$$F_\theta(t_i) = \frac{Phase(t_{i+1}) - Phase(t_i)}{2\pi * Fs}$$

where *Fs* is the sampling frequency.

Instantaneous theta amplitude was defined as the module of the LFP Hilbert transform and normalized by the mean LFP theta amplitude. The Pearson correlation coefficient was then calculated between the speed of the animal and theta frequency/amplitude.

A theta peak detection method was also used to calculate the instantaneous theta frequency. Theta peaks were detected with zero crossing of the instantaneous LFP phase and frequency was

deduced from the time between two successive theta peaks. This value was affected to all the time stamp of the corresponding cycle.

### Theta timescale correlation

To calculate the theta timescale lag between the spikes of two overlapping place fields, two cross-correlograms (CCGs) were computed (*Dragoi and Buzsáki, 2006*; *Robbe and Buzsáki, 2009*; *Skaggs et al., 1996*; CCGHeart; http://fmatoolbox.sourceforge.net/). First a 'Real-time scale' CCG was computed with a 1 s time window and 3 ms time bin. The CCG time lag was defined as the peak of the filtered CCG between [0–2] Hz. 'Theta time-scale' CCG was computed with a 200 ms time window and 1 ms time bin. The theta time lag was defined as the peak of the filtered CCG between [0–20] Hz. Only pairs of cells with a CCG mean bin count of 1 count/ms were included in this analysis. The relation between the CCG time lag and theta time lag was assessed using Pearson correlation.

### Preferred theta phase

Preferred theta phase and Mean Resultant Vector Length of each cell were defined thanks to circ_-mean and circ_r circular statistics MATLAB toolbox functions (*Berens, 2009*; https://github.com/circ-stat/circstat-matlab). Global phase 180° was defined as the maximal pyramidal cells activity (*Skaggs et al., 1996*).

### Statistics

All statistical analyses were conducted using MATLAB codes (MathWorks). For each distribution, a Lilliefors goodness-of-fit test was used to verify if the data were normally distributed and a Levene test was used to assess for equal variance. If normality or equal variance were not verified, we used the Wilcoxon rank sum test otherwise the Student t-test was used to compare two distributions. In case of multiple comparisons, the Kruskal-Wallis test with Bonferroni post-hoc test was used. Spatial correlations were computed using Pearson's correlation coefficient. Chi-square test was used to compare percentages of phase precessing cells. For circular distributions comparison, we first tested if they came from a Von-Mises distributions (Watson Test) with a common concentration (circ_ktest), if the distribution respected these constrains circular ANOVA: Watson-Williams multi-sample test for equal means (circ_wwtest) was applied.

## Acknowledgements

The authors thank Caroline Filippi for help with histology; Mathieu Pasquet, Ludovic Petit, Susanne Reichinnek and Robert Martinez for technical assistance; David Dupret, Pierre-Pascal Lenck-Santini, Vincent Hok, Francesca Sargolini, Michaël Zugaro and members of the Epsztein lab for useful discussions; Bryan Souza and Adriano Tort lab for comments on a preprint version of this article; the animal facility, administrative and imaging platforms of INMED for support. This study was supported by INSERM, a rising star grant from of the A*MIDEX project (n° ANR-11-IDEX-0001–02) funded by the «Investissements d'Avenir» French Government program (to JE), by the European Research Council under the European Community's Seventh Framework Program (ERC-2013-StG-338141_Intraspace to JE and ERC-2013-CoG-615699_NeuroKinematics to DR) and by the 'Agence National de la Recherche' (ANRJCJC to JK).

## Additional information

### Funding

| Funder | Grant reference number | Author |
| --- | --- | --- |
| Seventh Framework Programme | ERC-2013-CoG 615699-Neurokinematics | David Robbe |
| Agence Nationale de la Recherche | ANR-17-CE37-0005-GRIDSPACES | Julie Koenig |

| Seventh Framework Programme | ERC-2013-StG 338141 - IntraSpace | Jerome Epsztein |
|---|---|---|
| A*MIDEX under Investissements d'Avenir framework | ANR-11-IDEX-0001–02 | Jerome Epsztein |
| Région PACA | | Jerome Epsztein |
| Institut National de la Santé et de la Recherche Médicale | | Jerome Epsztein |

The funders had no role in study design, data collection and interpretation, or the decision to submit the work for publication.

### Author contributions

Romain Bourboulou, Conceptualization, Software, Formal analysis, Investigation, Visualization, Writing—review and editing; Geoffrey Marti, Conceptualization, Data curation, Software, Formal analysis, Visualization, Writing—review and editing; François-Xavier Michon, Software, Writing—review and editing; Elissa El Feghaly, Investigation, Writing—review and editing; Morgane Nouguier, Conceptualization, Investigation; David Robbe, Resources, Writing—review and editing; Julie Koenig, Conceptualization, Resources, Software, Formal analysis, Supervision, Investigation, Writing—review and editing; Jerome Epsztein, Conceptualization, Resources, Supervision, Funding acquisition, Visualization, Writing—original draft, Project administration, Writing—review and editing

### Author ORCIDs

Romain Bourboulou (iD) http://orcid.org/0000-0002-9133-8386
David Robbe (iD) http://orcid.org/0000-0002-9450-0553
Julie Koenig (iD) http://orcid.org/0000-0003-0516-6627
Jerome Epsztein (iD) http://orcid.org/0000-0002-5344-3986

### Ethics

Animal experimentation: All experiments were approved by the Institut National de la Santé et de la Recherche Médicale (INSERM) animal care and use committee and authorized by the Ministère de l'Education Nationale de l'Enseignement Supérieur et de la Recherche (agreement number 02048.02), in accordance with the European community council directives (2010/63/UE).

### Decision letter and Author response

Decision letter https://doi.org/10.7554/eLife.44487.030
Author response https://doi.org/10.7554/eLife.44487.031

## Additional files

### Supplementary files

• Supplementary file 1. Details of the recorded units and animal behavior per recording session in different mazes. TA: Track Active. Rw.: Number of rewards.
DOI: https://doi.org/10.7554/eLife.44487.026

• Supplementary file 2. Details of place cells' firing properties per recording session in different mazes.
DOI: https://doi.org/10.7554/eLife.44487.027

• Transparent reporting form
DOI: https://doi.org/10.7554/eLife.44487.028

### Data availability

All data generated or analyzed during this study are included in the manuscript and supporting files. Source data are provided for Figures 1–7.

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
