## [Decision Letter]

[Editors’ note: this article was originally rejected after discussions between the reviewers, but the authors were invited to resubmit after an appeal against the decision.]

Thank you for submitting your work entitled "Dynamic control of hippocampal spatial coding resolution by local visual cues" for consideration by *eLife*. Your article has been reviewed by a Senior Editor, a Reviewing Editor, and three reviewers. The following individuals involved in review of your submission have agreed to reveal their identity: Sara N Burke (Reviewer #1).

Our decision has been reached after consultation between the reviewers. Based on these discussions and the individual reviews below, we regret to inform you that your work will not be considered further for publication in *eLife*. Although potentially interesting, the results are not strong enough to support publication in *eLife*. Their novelty compared to previous results is not completely compelling, the interpretation is not entirely clear (e.g. presence of objects compared to enriched optic flow), and most importantly the results seem preliminary/ statistically weak in coming from very small numbers of animals in some analyses.

Reviewer #1:

The manuscript, "Dynamic control of hippocampal mapping resolution by local visual cues" by Bourboulou et al., documents the modulation of spatial coding by virtual 3-dimensional objects with a comprehensive analysis of firing properties of CA1 neurons in conditions with and without virtual objects. I have a few issues with the presentation of these data that would be helpful for the authors to address. First, throughout the Introduction and stated in the abstract the authors contend that, "whether hippocampal spatial coding resolution can be dynamically controlled within and between environments is unknown." This statement is false. In fact, there have been a number of studies showing that spatial coding resolution can be affected by behavioral/experimental parameters such as objects and active versus passive movement. Please see, Lee et al., 2012, Song et al., 2005, Terrazas et al., 2005 and Burke et al., 2011. The authors cite the Burke et al. (2011) paper but make no mention that this study also reported a decrease in place field size in the presence of objects, as well as an increase in the rate of theta phase precession. As it stands in the current presentation, the authors are upselling the novelty of their data.

A second major issue is how the statistics were conducted. It appears that cell number or place field number were the degrees of freedom for most analyses. Because multiple cells are recorded from the same animals, this is a nested design. In other words, multiple observations (cells) from a single subject (mouse) are treated like independent samples. The fact that many of the cells are from a common animal violates the statistical assumption that the observations are indeed independent. I refer the authors to Aarts et al., (2014) for an elegant description and meta-analysis of how such an approach can increase the chance of a Type I error. The data should be re-analyzed to account for the nested design of the experiment. Moreover, the authors do not report how many cells/fields were recorded from each mouse; so different animals could be making disproportionate contributions to the data.

Finally, the authors' use of the Hilbert transform to calculate theta phase is somewhat problematic. Hilbert imposes symmetry on the oscillation and it is known that theta is not symmetrical (Belluscio et al., 2012). This could lead to estimation errors for instantaneous phase that would obscure the quantification. At a minimum, the authors need to show that the shape of the theta oscillation (that is, the degree of asymmetry) did not vary between the object and no object conditions.

*Reviewer #2:*

In this paper, Bourboulou et al., show that the resolution of the hippocampal map is improved in the presence of 3D objects in the environment. They record hippocampal activity from head fixed mice running on virtual linear tracks with or without objects to show an improvement in spatial resolution and stability as well as improved theta phase precession.

While there are multiple major concerns with this paper, the biggest concern is the small number of experimental subjects per condition (**2**-3). More sessions per subject or more neurons per session cannot compensate for the possibility that the differences they saw could be the idiosyncrasies of individual subjects.

Major concerns:

1) Subsection “Low proportion of landmark vector cells in OT” "Because LV cells tend to systematically discharge near objects, these cells should discharge near the same object (s) in both back and forth trials."

This is not an accepted criterion to call a cell a LV cell. Both Deshmukh and Knierim as well as Geiller et al., papers, quoted by the authors, use the tendency to fire at multiple locations defined with respect to multiple objects to identify LV cells. The criteria in the present study merely identify bidirectional neurons. We know place fields tend to be unidirectional on 1D tracks. There's no reason why there can't be unidirectional LV cells on 1D tracks. Deshmukh and Knierim used 2D arenas, while Geuller et al., had 1D arenas, but unidirectional movement; neither of these studies have any information about bidirectionality of LV cells on 1D tracks.

Geiller et al., do refer to bidirectional predictive cells, "Indeed, it is worth noting that in a study 43 where local cues were laid on a linear track, place cells similar to LV cells were reported in significantly large numbers. These cells had bidirectional place fields that encoded in each direction an equidistant position ahead of a landmark, and were suggested to reflect view-invariant object information.", but are careful not to call these cells LV cells. In fact, the LV cell model by McNaughton et al., (1995), and the Collett et al., (1986) observations that model was supposed to explain would not predict this activity, since the animals (and the LV cells in the model) need to keep track of allocentric direction as well as distance from the landmark.

It is possible that the LV cells exhibit this bidirectional behavior in linear tracks, but there needs to be a comparison of behavior of LV cells in 2D and 1D before this bidirectional behavior gets labelled as LV.

Furthermore, this may not be the only possible representation of LV cell activity. Do the authors notice place fields equidistant from two or more objects in the same cell more often than expected by chance? That is the more classic LV cell behavior reported in Deshmukh and Knierim as well as Geiller et al., papers.

Subsection “Low proportion of landmark vector cells in OT” "In the track with objects, LV represented only 6.79% of all place cells. This corresponds to the proportion of LV cells recorded in area CA1 in the presence of real objects (Deshmukh and Knierim, 2013)."

This is an incorrect characterization of Deshmukh and Knierim results. McNaughton et al., (1995) posted that place cell vectors could be bound to one or more landmarks ("typically one, occasionally two, rarely more than two"). The percentage reported in Deshmukh and Knierim is that of LV cells with vectors bound to two or more landmarks; the paper had no means to characterize LV cells bound to a single land marks (which would be virtually indistinguishable from place cells with single place fields in absence of object manipulation). Thus, the low proportion of LV cells reported is the limitation of the method, not the actual proportion of LV cells, which is expected to be much higher.

Discussion section "the lateral entorhinal cortex where LV cells were first discovered (Deshmukh and Knierim, 2011)".

Deshmukh and Knierim (2011) did not report LV cells in LEC.

2) The papers switches between parametric and nonparametric tests, based on whether the data were normally distributed and had equal variance. While this is acceptable practice for individual tests, it is impossible to compare statistical significance across different comparisons of same quantities in the paper if one uses parametric tests while the other uses nonparametric tests. It will be better to use nonparametric tests throughout. In addition, median and range need to be reported when using nonparametric stats; mean and SEM reported in the paper are inappropriate. Conversely, reporting medians and range is inappropriate when performing parametric statistics (e.g. the box plots in Figure 1E).

3) Subsection “Effects of local visual cues on spatial coding resolution2: "similar rate of reward collections (OT: 1.70} 0.29 rewards/minute, n = 9 recording sessions in 3 mice; OT: 1.15} 0.09 rewards/minute" and "average running speed (OT: 14.1} 2.12 cm/s, n = 9 recording sessions in 3 mice; OT: 16.8} 1.58 cm/s, n = 5 recording sessions in 2 mice"

How is the average running speed for the without object track lower than that for OT (with object track), while the average reward rate is lower for OT? Do the average speed calculations exclude stationary periods? Do the OT mice sit longer at reward? More importantly, do they slow down at objects?

If they do slow down at objects, can this explain better spatial resolution/stability/place field dispersion in neural code near objects? i.e., can this be simply explained by slower speeds or longer time spent, ensuring better sampling of space near objects, and thus more (and less variable if the speeds at other locations vary more than those near objects) firing rate estimates at these locations than locations away from objects?

4) Subsection “Effects of local visual cues on spatial coding resolution” "There was a tendency for place field width (calculated on complete fields) to be lower in the track with objects (OT: 111 51.5} 3.33 cm, **n = 15 place fields**;".

This is a very small number of place fields (15) to be compared quantitatively. This ties in with the issue of small sample size (number of mice) used throughout the paper. Curiously, the authors report a greater number of place fields in the same without object condition elsewhere: subsection “Effects of local visual cues on spatial coding resolution” "Accordingly, spatial information (in bit/spike), a measure independent of place fields' detection (Skaggs et al., 1993) was very low in the track without object (0.06} 0.01 bit/spike, n = 48 place cells)". Place cells are defined as cells with at least 1 place field (subsection “Effects of local visual cues on spatial coding resolution”). Is that because most of these 48 cells don't meet the criterion of "complete place field" for even 1 field? Doesn't this make the definition of a "place cell" a bit too permissive? Clearly 31 of these fields were good enough for end track vs on track comparison (subsection “Virtual 3D objects improve spatial resolution locally”).

5) Subsection “Local visual cues improve hippocampal population coding accuracy” "We used the spike trains from all pyramidal 192 cells recorded (i.e., both the spatially modulated and nonspatially modulated cells) and compared decoded positions with actual positions of the animal in the virtual linear tracks."

Does using only the spatially modulated cells improve decoder accuracy? An explicit comparison of decoding accuracy using a matched number of spatially modulated cells is crucial, in addition to the "active cell" ensemble data presented here.

In continuation of the above point, subsection “Local visual cues improve hippocampal population coding accuracy”: "In both cases, downsampling was performed to equalize the number of cells used for decoding between the two conditions (20 active cells)."

Even after downsampling to 20 cells, most cells are place cells for with object but not place cells for without object condition. Matched number of place cells will complement this analysis.

6) The paper has no data to prove that it is the 3D nature of objects rather than their localized sensory information that is responsible for improvement in spatial representation.

7) Subsection “Recording procedure” "**On the day before recording**, animals were anesthetized (induction: isoflurane 3%; maintenance: Xylazine/Ketamine 10/100 mg/Kg supplemented with Buprenorphine 0.1 mg/Kg) and a craniotomy was drilled above one hippocampus (centered on a location -2 mm posterior and} 2.1 mm lateral from bregma)."

These lines and the rest of the paragraph in the methods give an impression that there was a single acute recording session per animal, the it is clear from the results that there were multiple recording sessions per animal (e.g. subsection “Effects of local visual cues on spatial coding resolution” "n = 5 recording sessions in 2 mice").

Each of these sessions included exposure to with and without object conditions: Subsection “Recording procedure” "All mice (n = 8) experienced first the familiar environment (either OT, OT or EOT) for around 20 back and forth trials. For mice trained in OT or OT (n = 3 and 2, respectively), this first exploration was followed, after 3 minutes of free running with the screens displaying a black background, by exploration of a new environment, identical to the previous one except for the presence of the three 3D objects (objects were added for mice trained in OT and removed for mice trained in OT) for another 20 consecutive back and forth trials." This means that the later sessions (session 2 onwards) had previous exposure to the "novel" condition; was there an effect of increasing familiarity on the neural response?

It is not clear from the description if the probes were fixed in one position on the first day of recording and reused over multiple days, or if they were inserted at different locations on different days. If they were at the same location, the statistics will be affected by the inflated degrees of freedom while recording from the same (or significantly overlapping) set of neurons over multiple days.

8) Discussion section "Nevertheless, End-track fields had a low spatial information content and stability when compared to fields recorded in OT (but similar to On-track fields recorded in the same maze). This argues against increased spatial coding resolution at these locations and further suggests a possible dissociation between overrepresentation and increased spatial coding resolution."

Or, it could simply be explained by the confusion caused by dissociation between the animal's movement and the arena caused by "teleportation" at the ends of the track.

9) Subsection "Effects of virtual objects in a visually enriched environment": This section lacks an essential control with enriched environment without objects.

10) Subsection “Virtual reality environments” "Outside the maze walls, two large 3D columns were positioned on each side (dimensions 8 x 8 x 47 cm; positions 58 and 143 cm from end wall) to provide additional visual cues.

While 58cm column position is close to an object, 143 cm position is not; there appears to be an enhancement in local stability near 143cm. But these columns are 3D, so doesn't detract from the overall analysis – just compounds the interpretation of this specific experiment. Is the reported improvement in neural code really due to an enrichment or merely an increase in number of discrete landmarks available to the animals?

*Reviewer #3:*

In this paper, the authors record hippocampal neurons as mice explore virtual reality environments that vary in their presence or absence of visual objects. They find that the resolution, defined as the number, spatial stability and scale, or place cells increases in the presence of visual objects. They go on to show that visual objects also enhance temporal coding, with theta phase precession emerging to a larger degree in the object rich compared to object poor environment. These results have interesting implications for our understanding of how hippocampal circuits dynamically change their coding properties based on the visual features available to the animal. In general, the experiments and the analyses in this paper are rigorously performed. The authors convincingly demonstrate a number of coding differences in hippocampal activity between the two environments. However, I do have concerns regarding interpretation, controls and sample size.

1) Interpretation and controls: I'm not certain to what degree the increased place cell resolution is driven by an 'object' per say versus the availability of improved optic flow sources. The use of a visually rich track with objects goes part way to addressing this issue but does not account for the fact that the optic flow from objects may carry more information than the optic flow from the walls, due to the proximity of the objects to the mouse. Moreover, the appropriate control here would be to record in the visually rich environment in the absence of the objects. It is possible that the presence of the objects induces a ceiling effect (perhaps coding cannot be further improved). More convincing to me would be that the visually rich environment *without* objects did not improve coding to the same degree as the presence of objects.

2) Sample size: Unless I misunderstood something, the number of place cells out of the total number of cells seems surprisingly low (48 and 103 out of 1124), which is concerning. I also have some concern about the number of mice used in the OT track (n = 2); while the overall cell number is large, I worry that this sample size is too small in terms of individual animals. Moreover, the session number is also rather small in some cases.

[Editors’ note: what now follows is the decision letter after the authors submitted an appeal.]

Thank you again for choosing to send your work entitled "Dynamic control of hippocampal spatial coding resolution by local visual cues" for consideration at *eLife*. Your article and your letter of appeal have been considered by a Senior Editor, a Reviewing Editor, and the original reviewers. We would happy to consider a new submission along the lines of your appeal, but please take note of the specific points below:

- Specifically, the authors would need to include the controls in a revised version, as well as the additional animal. The authors should redo the statistics taking into account the nested design after adding more data as indicated.

- In addition, the authors also need to substantially overhaul the writing of the entire manuscript to reflect the claims they make in the rebuttal/appeal.

For example, they state in the rebuttal, "We classified a cell as a LV cell if it responded either to multiple objects (having place field in the same object zones in back and forth trial) or to a single object (if they in addition code for that object's identity)." The entire LV cells section (subsection “Low proportion of landmark vector cells in OT”) does not mention anything about object identity being used for LV detection. Neither does the LV cells section (subsection “Landmark Vector cells detection”). The results do mention responding to the same object in both direction as a criterion, but that is not object identity, as a neuron firing bidirectionally for all objects will also be classified as LV cell.

- The resubmission will have to also deal with related issues, like the claims about percentages of cells that are landmark vector cells and how they compare with the other papers, definition of LV cells as bidirectional cells without confirming their LV nature in 2D environments etc.

---

## [Author Response]

[Editors’ note: the author responses to the first round of peer review follow.][…] Reviewer #1:The manuscript, "Dynamic control of hippocampal mapping resolution by local visual cues" by Bourboulou et al., documents the modulation of spatial coding by virtual 3-dimensional objects with a comprehensive analysis of firing properties of CA1 neurons in conditions with and without virtual objects. I have a few issues with the presentation of these data that would be helpful for the authors to address. First, throughout the Introduction and stated in the abstract the authors contend that, "whether hippocampal spatial coding resolution can be dynamically controlled within and between environments is unknown." This statement is false. In fact, there have been a number of studies showing that spatial coding resolution can be affected by behavioral/experimental parameters such as objects and active versus passive movement. Please see, Lee et al., 2012, Song et al., 2005, Terrazas et al., 2005 and Burke et al., 2011. The authors cite the Burke et al. (2011) paper but make no mention that this study also reported a decrease in place field size in the presence of objects, as well as an increase in the rate of theta phase precession. As it stands in the current presentation, the authors are upselling the novelty of their data.

We apologize for not having been clear enough when writing "whether hippocampal spatial coding resolution can be dynamically controlled within and between environments is unknown”. We are aware of the many papers that have shown that spatial coding can vary both quantitatively and qualitatively depending on different behavioral/experimental conditions. For example, place cells coding degrades when animals are exploring an environment in the absence of visual inputs (Lee et al., 2012) or in the absence of locomotion in a toy car (Terrazas et al., 2005; Song et al., 2005). But these are global changes in the absence of important sensory modalities. On the other hand, quantitative changes in place fields number, such as those associated with rewards or objects, are often interpreted as the coding of non-spatial information (see recent paper in Neuron by the Tank lab) and whether they could also correspond to increased spatial coding resolution remains unclear. We acknowledge that this sentence was ambiguous. We have modified this sentence in the Abstract and Introduction to clarify the fact that our main question is whether spatial coding resolution can quickly adapt to local features of a given environment with comparable sets of sensory cues available. To answer this question, we analyzed place cells coding both quantitatively and qualitatively. We show that spatial coding resolution can be increased both quantitatively and qualitatively locally around local visual cues. Using a Bayesian decoding framework we further show that spatial decoding accuracy, at the population level, is indeed better in these locations. Such local modulation of spatial coding resolution has not been reported before to the best of our knowledge and could have important implications for large scale navigation.

In the revised version of the manuscript we go deeper into the understanding of which features of local visual cues are important for this modulation by comparing the effect of 3D virtual objects with that of 2D patterns. We also show that several aspects of the temporal coding of spatial information were also modified by local sensory cues such as theta phase precession (which is close to 0 in the absence of local sensory cues) and theta timescale spike coordination. Altogether our results complement previous studies by showing that quantitative and qualitative changes in place cell coding can be co-modulated to locally set hippocampal spatial coding resolution.

Regarding our incomplete description of the Burke et al., 2011 study we want to first apologize for this omission. We did not specifically discussed changes in place field size in the previous version of the manuscript and therefore this result from the Burke et al., 2011 study was not highlighted. We now included a new paragraph in the Discussion section to discuss this point and notably discuss the fact that in the Burke et al., 2011 study decreased place field size was compensated for by increased place fields number such that the probability of place cell activation for any point in space was similarly low between the objects and non-objects conditions. However, because objects were distributed all along the track in this study their local effect on spatial coding resolution was not investigated. In our case however, despite a local reduction in place field size, the strong increase in the number of place fields in OZ resulted in a strong increase in the probability of place cell activation at these locations (OZ: 6.56 ± 0.32% /10 cm, n = 12 spatial bins of 10 cm, 6 in each direction; ØZ: 4.50 ± 0.29% /10 cm, n = 20 spatial bins of 10 cm, 10 in each direction; t_30_ = -4.54, P < 10^4^, two-tailed unpaired t-test). This result shows that while there might be a general mechanism to maintain a constant and low proportion of place cells activated at each position notably between dorsal and ventral parts of the hippocampus (Skaggs and McNaughton, 1992; Maurer et al., 2006), spatial coding resolution can nevertheless be increased within a given hippocampal region depending on the local cues present in the environment.

Concerning the increased rate of theta phase precession in the presence of real objects reported in the Burke et al., 2011 paper, it was not associated with a significant change in the correlation between phase and position (their Figure 10C). Thus, it could result from a scaling of theta phase precession rate with place field size (Huxter et al., 2003) allowing place field size-independent position coding. We agree that this point deserves some discussion which is now included in the revised version of the manuscript (Discussion section). In our study, theta phase precession was measured on normalized place field size, using only trials with a good spatial correlation with the mean place field in ØT (to also compensate for increased spatial dispersion in this condition) and the correlation between phase and position was significantly increased in not significantly different from 0 in the absence of local visual cues consistent with poor temporal coding of spatial information. Additionally, we assessed theta phase precession using place fields detected in single laps (which have comparable sizes in OT and ØT) and using comparisons between the frequency of theta rhythmicity in spike autocorrelograms and the frequency of the theta oscillation recorded in the LFP, a measure independent from place field detection (Geisler et al., 2007). The role of local visual cues in improving the temporal coding of spatial information is thus a genuine and novel finding of the present work.

A second major issue is how the statistics were conducted. It appears that cell number or place field number were the degrees of freedom for most analyses. Because multiple cells are recorded from the same animals, this is a nested design. In other words, multiple observations (cells) from a single subject (mouse) are treated like independent samples. The fact that many of the cells are from a common animal violates the statistical assumption that the observations are indeed independent. I refer the authors to Aarts et al., (2014) for an elegant description and meta-analysis of how such an approach can increase the chance of a Type I error. The data should be re-analyzed to account for the nested design of the experiment. Moreover, the authors do not report how many cells/fields were recorded from each mouse, so different animals could be making disproportionate contributions to the data.

The reviewer noted the nested design of our experiment where multiple cells were recorded simultaneously on the same day in the same animal during a recording session. We would like to highlight that reducing the number of animals was a recommendation of our local ethical committee. Nevertheless, this design could influence the result of our statistical tests. To control for the possible impact of this nested design on our statistics we re-ran our analysis using sessions number instead of cells or place fields number as the degree of freedom. We believe that session is the appropriate degree of freedom for these analyses because it removes the main dependency between cells recorded together on the same day in the same animal. Indeed, as now mentioned in the Results section, recordings were targeted to different locations on different recording sessions to avoid recording from the same cells (subsection “Recording procedure”). Using session number nevertheless allows a sufficiently high number of observations (n = 6-9) to avoid Type II errors, which could result from over downsampling the dataset(Aarts et al., 2014). Using sessions as the degree of freedom all main results were significant such as an increase in the proportion of place cells, a lower place field dispersion, an increased place field stability, decreased out-of-field vs in-field firing ratio and increased spatial information content in the presence of objects. All local effects of objects were also significant. This new analysis further strengthens our results showing a strong effect of 3D visual cues on hippocampal spatial coding resolution.

Finally, the authors' use of the Hilbert transform to calculate theta phase is somewhat problematic. Hilbert imposes symmetry on the oscillation and it is known that theta is not symmetrical (Belluscio et al., 2012). This could lead to estimation errors for instantaneous phase that would obscure the quantification. At a minimum, the authors need to show that the shape of the theta oscillation (that is, the degree of asymmetry) did not vary between the object and no object conditions.

To take into account possible pitfall linked to Hilbert transform, we re-ran our theta phase precession analysis using a waveform-based theta-phase estimation (which takes into account theta asymmetry; Belluscio et al., 2012; Fernandez-Ruiz et al., 2017). More specifically, the broadband (1 to 60 Hz) filtered signal was used to find peaks and troughs of each theta cycle. The intervals in which to detect extrema were determined using zerocrossings of the narrowband-filtered (4-10 Hz) signal. Peaks (0°) of theta waves were identified as local maxima and troughs (180°) as local minima. Descending points (90°) were identified as the zero crossings of the signal between the trough and peak, and ascending points (270°) were identified as the zero crossings of the signal between peak and trough. The waveform-based theta phase was obtained by interpolating phase values between these phase quadrants. The results were identical to previous analysis using Hilbert transform as indicated in the text (subsection “Effects of local cues on hippocampal temporal coding resolution”) and in new Figure 7—figure supplement 3. We also want to highlight that in the original version of the paper the difference in theta phase precession between ØT and OT conditions was also detected as a higher frequency of theta modulated spikes compared to the frequency of the field theta oscillation (Geisler et al., 2013) in OT but not in ØT. This measure does not depend on the symmetry of the LFP theta oscillation. All these measures indicated altered theta phase precession in the absence of local visual cues. The effect of object on temporal coding is also observed for theta timescale spike coordination which does not depend on the shape of the LFP theta oscillation.

Reviewer #2:[…] While there are multiple major concerns with this paper, the biggest concern is the small number of experimental subjects per condition (**2**-3). More sessions per subject or more neurons per session cannot compensate for the possibility that the differences they saw could be the idiosyncrasies of individual subjects.

We agree that the number of animals used for comparisons between the familiar conditions in the previous version of the manuscript was small but we want to stress out that all animals (n = 5) experienced both the track with objects and the track without objects with varying degrees of familiarity. Our new analysis focusing on spatial coding resolution during the first session in the new condition (see below) shows that familiarity is not required for the effect of objects on spatial coding resolution. Because all animals experienced both conditions it is unlikely that the difference observed between familiar conditions are the result of idiosyncratic differences between our animals. For example, the same cell ensemble recorded from the same animals show good spatial coding resolution in fam OT and poor spatial coding resolution in new ØT and vice versa. In order to nevertheless address this point, we trained a new animal to run in the familiar track with objects (OT, the familiar condition where data from only two mice were included in the previous version of the manuscript). This mouse was recorded during 3 sessions in the familiar condition and 3 sessions in the new condition (after object removal). Recording from this mouse yielded 129 additional active cells and 90 additional place cells to the dataset. All previous results were confirmed when data from this mouse were included. The only difference was observed when comparing the maze with object and the enriched maze with objects. We observed that further enriching the environment with additional 3D columns and patterns could further decrease out-of-field versus in-field firing ratio and increase inter-trial stability while all other effects remained non-significant.

Major concerns:1) Subsection “Low proportion of landmark vector cells in OT” "Because LV cells tend to systematically discharge near objects, these cells should discharge near the same object (s) in both back and forth trials." […]Discussion section "the lateral entorhinal cortex where LV cells were first discovered (Deshmukh and Knierim, 2011)".Deshmukh and Knierim (2011) did not report LV cells in LEC.

We agree with the reviewer that our definition of Landmark Vector cells does not exactly match the definition proposed by the reviewer and other authors and we apologize for the confusion. Following the reviewer comment we believe that it is more appropriate to call these cells objects responsive (OR) cells, which have been shown in the hippocampus to have single or multiple fields near objects (Deshmukh and Knierim, 2013). Indeed, these cells have a local modulation close to objects and are more likely to explain our results. We rewrote this part of the manuscript to clarify the fact that we do not only look for OR cells responding to a single object but also for cells responding to multiple objects. Indeed, we defined individual object zones (IOZ) around each object and to be classified as an OR cell a place cell should have at least one place field in the same IOZ in back and forth trials (but could have multiple place fields in multiple identical IOZs in back and forth trials). In both cases, because the same objects are encountered in both back and forth trials OR cells will be bidirectional. Conversely, any bidirectional place cell will not automatically be identified as an OR cell. For example, if a cell has two fields, one in each direction, but these fields are not located in the same IOZ for back and forth trial then this cell will not be considered as an OR cell. We agree with the reviewer that we cannot speculate on the bi-directional behavior of LV cells, but we think that an OR cell should respond to the same object in both back and forth trials. We added a paragraph in the Results section to insist on this point. Using this analysis we found that OR cells corresponded to 2.07% of the place cells we recorded and thus cannot quantitatively explain the results we observed.

2) The papers switches between parametric and nonparametric tests, based on whether the data were normally distributed and had equal variance. While this is acceptable practice for individual tests, it is impossible to compare statistical significance across different comparisons of same quantities in the paper if one uses parametric tests while the other uses nonparametric tests. It will be better to use nonparametric tests throughout. In addition, median and range need to be reported when using nonparametric stats; mean and SEM reported in the paper are inappropriate. Conversely, reporting medians and range is inappropriate when performing parametric statistics (e.g. the box plots in Figure 1E).

We used parametric test and nonparametric tests appropriately in the paper based on whether the data were normally distributed and had equal variance as noted by the reviewer. Parametric tests are more sensitive than non parametric tests because they use data with more comparable distributions thus not using a parametric test when possible increases the risk of type II error which is the risk of not detecting a difference when the difference actually exists. We think that comparing statistical significance across different comparisons of the same quantities is complicated by the fact that even if the same quantity is compared the sample size is likely to be different thus precluding such comparisons. Concerning median vs mean, more and more journals encourage to show the results in the form of box plots to have a better idea of data distribution. However, reporting the mean and SEM is a common practice that allows comparisons between studies. We thus decided to report the mean + SEM in the text while median and distribution are visible in the box plots.

3) Subsection “Effects of local visual cues on spatial coding resolution2: "similar rate of reward collections (OT: 1.70} 0.29 rewards/minute, n = 9 recording sessions in 3 mice; OT: 1.15} 0.09 rewards/minute" and "average running speed (OT: 14.1} 2.12 cm/s, n = 9 recording sessions in 3 mice; OT: 16.8} 1.58 cm/s, n = 5 recording sessions in 2 mice".How is the average running speed for the without object track lower than that for OT (with object track), while the average reward rate is lower for OT? Do the average speed calculations exclude stationary periods? Do the OT mice sit longer at reward? More importantly, do they slow down at objects?If they do slow down at objects, can this explain better spatial resolution/stability/place field dispersion in neural code near objects? i.e., can this be simply explained by slower speeds or longer time spent, ensuring better sampling of space near objects, and thus more (and less variable if the speeds at other locations vary more than those near objects) firing rate estimates at these locations than locations away from objects?

The average speed calculation indeed excludes stationary periods (with a threshold of 2cm/s). We calculated the average time spent at reward locations in OT and ØT tracks and found no significant differences (ØT: 26.2 ± 4.44 seconds, n = 9 sessions in 3 mice; OT: 26.9 ± 4.7 seconds, n = 8 sessions in 3 mice; *Z* = -0.11, *P* = 0.91, two-tailed unpaired t test). To determine if mice were slowing down at objects we specifically compared average running speed in OZ vs ØZ. We found a trend for animals to slow down in OZ compared to ØZ but the difference was small and not significant (ØZ: 16.1 ± 1.43 cm/s, n = 8 sessions in 3 mice; OZ: 14.0 ± 1.06 cm/s, n = 8 sessions in 3 mice; *Z* = 1.2, *P* = 0.25, two-tailed unpaired t test).

We however do not think that this trend alone can explain our results because speed was on average lower in ØT compared with OT and spatial resolution was lower in this track compared to OT. Furthermore animals slowed down much less in OZ compared to ØZ in the new OT condition with object (ØZ: 12.2 ± 2.22 cm/s, n = 6 sessions in 3 mice; OZ: 11.5 ± 2.13 cm/s, n = 6 sessions in 3 mice; *Z* = 0.25, *P* = 0.81, two-tailed unpaired t test), but differences in coding quality between OZ and ØZ was identical to the one observed in the familiar OT condition (Figure 4—figure supplement 1).

4) Subsection “Effects of local visual cues on spatial coding resolution” "There was a tendency for place field width (calculated on complete fields) to be lower in the track with objects (OT: 111 51.5} 3.33 cm, **n = 15 place fields**;".This is a very small number of place fields (15) to be compared quantitatively. This ties in with the issue of small sample size (number of mice) used throughout the paper. Curiously, the authors report a greater number of place fields in the same without object condition elsewhere: subsection “Effects of local visual cues on spatial coding resolution” "Accordingly, spatial information (in bit/spike), a measure independent of place fields' detection (Skaggs et al., 1993) was very low in the track without object (0.06} 0.01 bit/spike, n = 48 place cells)". Place cells are defined as cells with at least 1 place field (subsection “Effects of local visual cues on spatial coding resolution”). Is that because most of these 48 cells don't meet the criterion of "complete place field" for even 1 field? Doesn't this make the definition of a "place cell" a bit too permissive? Clearly 31 of these fields were good enough for end track vs on track comparison (subsection “Virtual 3D objects improve spatial resolution locally”).

We agree that 15 is a small number of place fields to be compared quantitatively but we disagree with the fact that it is linked to a small sample size. We recorded a total of 526 putative pyramidal cells in this condition in three mice during nine recording sessions. However, a very low percentage of these cells were spatially modulated. Furthermore, most of these cells had end-track fields, which are usually not complete, and this is why the number of complete fields to be compared is low. We do not think that our criterion to detect place fields is too permissive because only 48 cells were overall detected as place cells out of 526 putative pyramidal cells in this condition. This detection is based on a shuffling procedure with additional place field length and stability requirements. The low number of place cells with complete place fields in ØT could explain why place field size was not globally significantly decreased in OT vs ØT. However, an important result of the present work is that spatial coding resolution can be locally modulated and when we compared place field size between OZ and ØZ with larger number of fields (77 and 80 respectively) we could see a significant decrease in OZ compared to ØZ. Altogether we think that this result clearly shows an effect of 3D objects on local place field size in dorsal hippocampus.

5) Subsection “Local visual cues improve hippocampal population coding accuracy” "We used the spike trains from all pyramidal 192 cells recorded (i.e., both the spatially modulated and nonspatially modulated cells) and compared decoded positions with actual positions of the animal in the virtual linear tracks."[…]Even after downsampling to 20 cells, most cells are place cells for with object but not place cells for without object condition. Matched number of place cells will complement this analysis.

Our Bayesian decoding analysis had two purposes. The first was to see the effect of changes in both place cell proportion and place cell coding quality between OT and ØT on position coding at the population level. The second was to see whether, despite the low proportion of spatially modulated cells in ØT, position coding could still be accurate due to the contribution of non-spatially modulated cells as recently observed in CA1 (Meshulam et al., 2017). We agree with the reviewer that using spatially modulated cells would improve decoding accuracy but the low proportion of spatially modulated cells in ØT is a clear result of our study and its impact on spatial coding at the population level cannot be ignored. On the other hand, we agree with the reviewer that trying to dissociate the impact of quantitative and qualitative changes in place cell coding on spatial coding at the population level is interesting. In an attempt to dissociate these effects, we performed a new analysis comparing decoding between the two conditions using a 3 fold higher number of cells in ØT compared to OT (to compensate for the fact that place cell proportion is 3 time lower in ØT compared to OT). Thus, we compared decoding with 15 cells in ØT and 5 cells in OT, 30 cells in ØT and 10 cells in OT and 45 cells in ØT and 15 cells in OT. In all cases decoding was significantly lower in ØT than in OT. This result show that place cell coding quality is probably as important as place cells number to accurately code for space.

6) The paper has no data to prove that it is the 3D nature of objects rather than their localized sensory information that is responsible for improvement in spatial representation.

We fully agree with the reviewer that this experiment was missing in the previous manuscript. In order to address this point, we trained two additional mice to run in a track enriched with different 2D patterns at different locations along the maze but devoided of the original 3D objects (pattern no object track or PØT). In this maze, the number of active cells was not different between the ØT and OT conditions. The proportion of place cells tended to increase compared to ØT but this effect did not reach significance. As a consequence, the proportion of place cells in PØT remained significantly lower than in OT. Thus, 2D local visual cues are not as efficient as 3D objects to quantitatively engage the hippocampal mapping system. Interestingly the distribution of place field was uniform in this maze showing that patterns indeed provided local visual cues important for place fields location. Qualitatively place field size was not different in PØT compared to ØT while place field spatial dispersion was significantly reduced to a level similar to the one observed in OT. For all other measures such as inter trial firing stability, out-of-field versus in-field firing ratio and spatial information, values in PØT were significantly higher than in ØT but significantly lower than in OT despite the fact that cues were distributed all along the maze in PØT while clustered in specific locations in OT.

We conclude that 2D local cues did improve spatial coding resolution but to a lower extend compared to 3D visual cues.

All these results are included in a new subsection “Effects of 2D wall patterns on hippocampal spatial coding resolution” and illustrated by a new Figure 5.

7) Subsection “Recording procedure” "**On the day before recording**, animals were anesthetized (induction: isoflurane 3%; maintenance: Xylazine/Ketamine 10/100 mg/Kg supplemented with Buprenorphine 0.1 mg/Kg) and a craniotomy was drilled above one hippocampus (centered on a location -2 mm posterior and} 2.1 mm lateral from bregma)."These lines and the rest of the paragraph in the methods give an impression that there was a single acute recording session per animal, the it is clear from the results that there were multiple recording sessions per animal (e.g. subsection “Effects of local visual cues on spatial coding resolution” "n = 5 recording sessions in 2 mice").

We agree that the description of the recording procedure in the former version of the manuscript was misleading. Notably it was unclear whether several recording sessions were performed per animal. As suggested by the reviewer, this part of the Materials and methods section has been rewritten to clarify this point: “On the day before the first recording session, animals were anesthetized (induction: isoflurane 3%; maintenance: Xylazine/Ketamine 10/100 mg/Kg supplemented with Buprenorphine 0.1 mg/Kg) and a craniotomy was drilled above one hippocampus (centered on a location -2 mm posterior and ± 2.1 mm lateral from bregma). The craniotomy was covered with agarose (2% in physiological saline) then sealed with silicon elastomer (Kwik-Cast, World Precision Instruments). This craniotomy was used to record acutely during 2-3 consecutive days (with the probe lowered in a new location every time). Then a second craniotomy was performed over the other hippocampus following the same procedure and recordings were performed during 2-3 additional days.” (subsection “Recording procedure”).

Each of these sessions included exposure to with and without object conditions: Subsection “Recording procedure” "All mice (n = 8) experienced first the familiar environment (either OT, OT or EOT) for around 20 back and forth trials. For mice trained in OT or OT (n = 3 and 2, respectively), this first exploration was followed, after 3 minutes of free running with the screens displaying a black background, by exploration of a new environment, identical to the previous one except for the presence of the three 3D objects (objects were added for mice trained in OT and removed for mice trained in OT) for another 20 consecutive back and forth trials." This means that the later sessions (session 2 onwards) had previous exposure to the "novel" condition; was there an effect of increasing familiarity on the neural response?

In our previous analysis, all sessions recorded in the new condition were pulled together despite the fact that the degree of familiarity increased between the first session and the last session in the new condition. To see if this familiarization as an effect on the changes observed we reran our analysis but comparing cells recorded in the familiar condition with cells recorded during the first session in the new environment. All differences in terms of proportion of place cells, out-/in-field firing ratio, spatial information, place field dispersion and stability were already significantly observed during this first session (as illustrated in new supplementary Figure 4—figure supplement 1). This result is now included in the manuscript (subsection “Fast dynamics of spatial coding resolution tuning upon objects manipulation”). We conclude that spatial coding resolution changes during object manipulation occur instantaneously without the need of familiarization in the new condition. This result is also in line with the fast kinetics revealed by the similarity map analysis between trials before and after the manipulation.

It is not clear from the description if the probes were fixed in one position on the first day of recording and reused over multiple days, or if they were inserted at different locations on different days. If they were at the same location, the statistics will be affected by the inflated degrees of freedom while recording from the same (or significantly overlapping) set of neurons over multiple days.

In order to avoid recording from the same neurons probes were inserted in different positions on different days as now stated in the Materials and methods section.

8) Discussion section "Nevertheless, End-track fields had a low spatial information content and stability when compared to fields recorded in OT (but similar to On-track fields recorded in the same maze). This argues against increased spatial coding resolution at these locations and further suggests a possible dissociation between overrepresentation and increased spatial coding resolution."Or, it could simply be explained by the confusion caused by dissociation between the animal's movement and the arena caused by "teleportation" at the ends of the track.

Because of reward zone exclusion, end-track field are located 20 cm away from the wall were rewards are delivered and teleportation occur. It is thus unlikely that teleportations are responsible for the low stability and information content of end track fields. Furthermore, the low spatial information content and stability of end track fields was similar to that of ontrack fields recorded in the same maze, which are away from teleportation zones. This result nevertheless shows that place field density can be increased at the end of the track without a concurrent increase in the quality of the recorded place cells.

9. Subsection "Effects of virtual objects in a visually enriched environment": This section lacks an essential control with enriched environment without objects.

We now included these data with the new track enriched in visual patterns but without 3D objects.

10) Subsection “Virtual reality environments” "Outside the maze walls, two large 3D columns were positioned on each side (dimensions 8 x 8 x 47 cm; positions 58 and 143 cm from end wall) to provide additional visual cues.While 58cm column position is close to an object, 143 cm position is not; there appears to be an enhancement in local stability near 143cm. But these columns are 3D, so doesn't detract from the overall analysis – just compounds the interpretation of this specific experiment. Is the reported improvement in neural code really due to an enrichment or merely an increase in number of discrete landmarks available to the animals?

In the enriched maze both 2D visual and 3D visual cues were added to decipher if the coding resolution could be further improved. We found no increase in spatial coding resolution in terms of the proportion of spatially modulated cells, place field dispersion and spatial information content however the stability index and spatial information content were significantly increased. As rightfully noted by the reviewer these improvements could result from the presence of additional 3D local cues (the 3D columns) in the environment. To answer this question, we compared the local stability in the object zone comprising two 3D objects in OT with the stability of the same object zone comprising three 3D objects in EOT and the stability was not significantly different (OT: 0.57 ± 0.02 vs EOT: 0.62 ± 0.02; *Z* = -1.31; *P* = 0.18; two-tailed WRS test). This result suggests that when 3D objects are already present in a location, increasing their number does not further increase spatial stability. However, in no-object zones adding a new 3D visual cue could increase local stability.

Reviewer #3:In this paper, the authors record hippocampal neurons as mice explore virtual reality environments that vary in their presence or absence of visual objects.[…] However, I do have concerns regarding interpretation, controls and sample size.

We thank the reviewer for positive comments on our study.

1) Interpretation and controls: I'm not certain to what degree the increased place cell resolution is drive by an 'object' per say versus the availability of improved optic flow sources. The use of a visually rich track with objects goes part way to addressing this issue but does not account for the fact that the optic flow from objects may carry more information than the optic flow from the walls, due to the proximity of the objects to the mouse. Moreover, the appropriate control here would be to record in the visually rich environment in the absence of the objects. It is possible that the presence of the objects induces a ceiling effect (perhaps coding cannot be further improved). More convincing to me would be that the visually rich environment without objects did not improve coding to the same degree as the presence of objects.

We agree with the reviewer that understanding the factors explaining the effects of visual objects on hippocampal spatial coding resolution would be of great interest. Increased optic flow is an interesting possibility. In order to address this point, we trained two additional mice to run in a track enriched with different 2D patterns at different locations along the maze but devoid of the original 3D objects (pattern no object track or PØT). In this maze, the number of active cells was not different from the ØT and OT conditions. The proportion of place cells tended to increase compared to ØT but this effect did not reach significance. On the other hand, the proportion of place cells in PØT remained significantly lower than in OT. Thus, 2D local visual cues are not as efficient as 3D objects to quantitatively engage the hippocampal mapping system. Interestingly, the distribution of place field was uniform in this maze showing that patterns indeed provided local visual cues important for place field location. Qualitatively, place field size was not different in PØT compared to ØT while place field spatial dispersion was significantly reduced to a level similar to the one observed in OT. For all other measures of place field quality such as intertrial firing stability, out-of-field versus in-field firing ratio and spatial information values in PØT were significantly higher than in ØT but significantly lower than in OT despite the fact that cues were distributed all along the maze in PØT while clustered in specific locations in OT. We conclude that 2D local cues did improve spatial coding resolution but to a lower extend compared to 3D visual cues. These results are described in a new paragraph in the RResults section and illustrated in a new Figure 5. We hope that the reviewer will be convinced that the 3D nature of visual object is responsible, at least partly, for the observed effect.

2) Sample size: Unless I misunderstood something, the number of place cells out of the total number of cells seems surprisingly low (48 and 103 out of 1124), which is concerning. I also have some concern about the number of mice used in the OT track (n = 2); while the overall cell number is large, I worry that this sample size is too small in terms of individual animals. Moreover, the session number is also rather small in some cases.

In the original version of the article 1124 referred to the total number of identified (clustered) cells recorded in the study which included both active and silent cells as well as interneurons recorded in all the tracks/sessions including the EOT track while 48 and 103 place cells referred to the number spatially modulated cells recorded in the familiar conditions in OT and ØT. So these numbers cannot be compared directly. In the revised version we mentioned only the number of neurons recorded in these conditions (OT and ØT) to make comparisons between numbers more straightforward. While the percentage of cells in ØT is very low (48 out of 526 putative pyramidal cells, 9%), as also reported before in rats navigating VR environment with only distal visual cues available (Ravassard et al., 2013), the percentage of place cells in OT (193/495 putative pyramidal cells, 39%) is more comparable to values reported in the literature for real environments in mice.

We agree with the reviewer that n=2 is a borderline small number of animals. To address this concern, in the revised version of the manuscript we included data from a new mouse which was trained in the OT maze (were only data from two mice were included before). This mouse was recorded during 3 training sessions which yielded a total of 129 additional active cells and 90 additional place cells. With this mouse included, 6 mice recorded during 17 sessions experienced both conditions with different degrees of familiarity. Our analysis of online manipulation (notably the new analysis focusing on the first session in the new environment see below and Figure 4—figure supplement 1) revealed that main effects of objects on spatial coding resolution are already observed during this first session. We are thus confident that the number of mice and sessions is now sufficient to hold the main conclusions of the paper on the local effect of 3D visual cues on spatial coding resolution.

[Editors’ note: the author responses to the re-review follow.]

[…] We would happy to consider a new submission along the lines of your appeal, but please take note of the specific points below:- Specifically, the authors would need to include the controls in a revised version, as well as the additional animal. The authors should redo the statistics taking into account the nested design after adding more data as indicated.- In addition, the authors also need to substantially overhaul the writing of the entire manuscript to reflect the claims they make in the rebuttal/appeal. […]- The resubmission will have to also deal with related issues, like the claims about percentages of cells that are landmark vector cells and how they compare with the other papers, definition of LV cells as bidirectional cells without confirming their LV nature in 2D environments etc.

One major concern raised by all reviewers was the low number of animals used in our study. To address this concern, we trained an additional mouse to run in the track with objects (one of the main conditions in the paper for which data from only two mice were reported in the previous version of the manuscript). This mouse was recorded during 3 sessions in the familiar condition and 3 sessions in the new condition (after object removal). Recording from this mouse yielded 129 additional active cells and 90 additional place cells. All previous results were confirmed when data from this mouse were included. The only difference was a significant decrease in out- versus in-field firing ration and a significant increase in stability when comparing the track with objects and the enriched track with objects. This difference does not contradict the main conclusions of the paper but further suggests that additionally enriching an environment with 3D columns and patterns can further improve some aspects of spatial coding resolution.

Another important point concerned the fact that we used a nested design in our experiments with several recording sessions per animal and several cells recorded per session. Although this design is common in studies performing acute recordings in headfixed animals we nevertheless tried to address this point. We reran our main analyses using sessions and not place fields as the main degree of freedom and all main results such as increased place field spatial and temporal stability decreased out of field firing and increased spatial information content remained statistically significant. Furthermore, local differences in place field proportion, size, stability and spatial information content between object zone and non-object zone were also significant when session was the degree of freedom used. We think that this analysis rules out a strong effect of the nested designed we used on our statistics.

Reviewer 2 and 3 were also concerned by the fact that there was no data showing that the 3D nature of objects was responsible for the effects we observed. To address this point, we trained two additional mice (n = 359 active cells, n = 157 place cells) to run in a track enriched with 2D visual patterns along the track but in the absence of 3D visual cues such as objects or columns. Enriching the track with these 2D cues had no significant effect on active cells nor place cells proportion and increased place cell coding accuracy but significantly less than 3D objects despite the fact that these cues were distributed all over the track unlike objects. Notably, out- vs in-field firing ratio was significantly higher in the presence of 2D patterns compared to 3D objects and spatial information and stability were significantly lower. Interestingly Bayesian decoding accuracy in the presence of these 2D cues was similar to the one observed in the non-object zone of the maze with object and significantly lower than that observed in the object zone. We thus conclude that the 3D nature of object plays an important role in the strong increase of spatial coding resolution observed.

To summarize in this revised version of the manuscript we:

1) Included an additional mouse in the track with object condition and reran all our analysis on global and local effects of 3D visual cues on place cells’ coding including Bayesian decoding, theta phase precession and theta timescale spike coordination. All results were confirmed with this additional mouse with the few exceptions mentioned above.

2) Re-ran our statistics using session number as the degree of freedom.

3) Included a new experimental condition with a track enriched with 2D patterns but devoid of 3D visual cues showing that 2D local visual cues could improve spatial coding but to a lower extend compared to 3D local cues (this is exactly the result predicted by reviewer 3).

4) Re-ran temporal coding analysis (i.e. theta phase precession) not using Hilbert transform but using waveform-based theta-phase estimation, an analysis which takes into account theta asymmetry and all results were consistent with our previous findings.

5) Clarified the LV cells section by requalifying these cells as object responsive (OR) cells (which are better suited as a possible explanation for our local effects close to the objects while LV cells can discharge away from landmark or objects).

6) Performed new analysis of behavior showing that animal speed was not significantly different in object versus non-object zones in the track with objects.

7) Performed new Bayesian decoding comparisons taking into account the lower proportion of place cells in the track without object. More specifically, because the proportion of place cells is ~ 3 times higher in OT compared to ØT we compared decoding using 5 cells in OT and 15 cells in ØT, 10 cells in OT with 30 cells in ØT, 15 cells in OT and 45 cells in ØT. In all cases decoding accuracy was significantly higher with cells recorded in OT compared to ØT.

8) Performed new analysis showing that the effects observed after object manipulations are already observed for the very first session in the new condition consistent with the fast dynamics of spatial coding resolution adaptation.